# Joint Design of Protein Surface and Structure Using a Diffusion Bridge Model

**Guanlue Li**
University of Hamburg
Hamburg, Germany
guanlue.li@uni-hamburg.de

**Xufeng Zhao**
University of Hamburg
Hamburg, Germany
xufeng.zhao@uni-hamburg.com

**Fang Wu**$^*$
Stanford University
Stanford, CA, USA
fangwu97@stanford.edu

**Sören Laue**$^*$
University of Hamburg
Hamburg, Germany
soeren.laue@uni-hamburg.de

## Abstract

Protein-protein interactions (PPIs) are governed by surface complementarity and hydrophobic interactions at protein interfaces. However, designing diverse and physically realistic protein structure and surfaces that precisely complement target receptors remains a significant challenge in computational protein design. In this work, we introduce PepBridge, a novel framework for the joint design of protein surface and structure that seamlessly integrates receptor surface geometry and biochemical properties. Starting with a receptor surface represented as a 3D point cloud, PepBridge generates complete protein structures through a multi-step process. First, it employs denoising diffusion bridge models (DDBMs) to map receptor surfaces to ligand surfaces. Next, a multi-model diffusion model predicts the corresponding structure, while Shape-Frame Matching Networks ensure alignment between surface geometry and backbone architecture. This integrated approach facilitates surface complementarity, conformational stability, and chemical feasibility. Extensive validation across diverse protein design scenarios demonstrates PepBridge's efficacy in generating structurally viable proteins, representing a significant advancement in the joint design of top-down protein structure. The code can be found at https://github.com/guanlueli/Pepbridge.

## 1 Introduction

Proteins are fundamental biological macromolecules that perform their functions through intricate interactions with other biomolecules, particularly through protein-protein interactions (PPIs) [22]. PPIs are primarily determined by surface complementarity and hydrophobic interactions at the interface regions, which facilitate specific and stable binding [36]. Understanding and designing PPIs is a central challenge in computational protein design, which seeks to predict sequences, generate structures, and design proteins with tailored properties while adhering to biochemical and geometric constraints [9]. These constraints are crucial for engineering proteins with desired binding characteristics and functional properties. Recent studies underscore that a protein's surface features, such as geometry and biochemical properties, have a more direct influence on its biological function than its sequence or backbone structure alone [20, 44, 53]. This insight is particularly relevant to PPIs, where interacting protein complexes exhibit geometric complementarity in the 3D space.

---

$^*$Corresponding authors.

39th Conference on Neural Information Processing Systems (NeurIPS 2025).

The interacting surfaces conform to their ligands' shapes and chemical properties, highlighting the importance of surface characteristics in protein design.

Protein design methods can generally be categorized into three approaches: sequence-based methods [15, 30, 49], structure-based methods [50, 55, 58], and sequence-structure co-design approaches [21, 25]. Sequence-based and structure-based methods focus on isolated aspects, which simplifies modeling but limits their ability to explore interactions at interface regions. Co-design approaches aim to holistically model both sequence and structure to capture their interdependence, yet they still struggle to accurately represent interface interactions. Providing the crucial role of protein surface analysis in predicting interaction sites and inferring PPIs [35, 46, 47], more efforts have considered comprising surface geometry and biochemical properties for protein discovery in parallel. For instance, Gainza et al. [14] built a surface-centric *de novo* design framework to capture the physical and chemical determinants of molecular recognition for new protein binders. Subsequent works [31] extract surface fingerprints from protein-ligand complexes for innovative drug-controlled cell-based therapies. Another line of works [44, 48] incorporates surface point clouds augmented with biochemical properties for protein engineering. Despite these advancements, existing methodologies face several limitations: **(i)** Limited ability to generate diverse yet receptor-compatible surface configurations. **(ii)** Lack of explicit modeling to establish robust correspondences between molecular shapes and backbone structures. **(iii)** Absence of a comprehensive strategy for top-down protein design, where coherent protein structures are generated based on receptor surface features.

To address these challenges, we introduce PepBridge, a novel framework for top-down protein design based on a multi-modal diffusion approach [17, 41–43]. As shown in Figure 1, given a receptor represented as a surface point cloud and structure annotated with geometric and biochemical properties, PepBridge generates a complete protein structure, including both the upper surface and the underlying residue structure. Notably, PepBridge leverages denoising diffusion bridge models (DDBMs) [62, 63], which interpolate between paired distributions, enabling the direct mapping of receptor surfaces to ligand surfaces while preserving physical and biochemical relevance. For structure generation, PepBridge employs an SE(3) diffusion model for backbone prediction, a torus diffusion model for torsion angle generation, and a logit-normal diffusion model for residue identity prediction. To ensure alignment and consistency, we introduce a Shape-Frame Matching Network that learns correspondences between generated ligand surfaces and backbone structures.

Our main contributions are as follows:

- **Unified Protein Design Framework**: We present PepBridge, a novel framework that jointly designs protein surfaces and structures by integrating receptor surface geometry and biochemical properties—tackling core challenges in top-down protein design.

- **Methodological Advances**: PepBridge incorporates DDBMs to generate receptor-compatible ligand surfaces and a multi-modal diffusion model for peptide structure prediction. Additionally, a Shape-Frame Matching Network is introduced to align generated surfaces and backbone structures, improving geometric and biochemical consistency.

- **Effective Validation**: We demonstrate the efficacy of PepBridge through extensive validation on peptide design tasks, showcasing its ability to generate diverse, structurally viable proteins with receptor-specific binding characteristics.

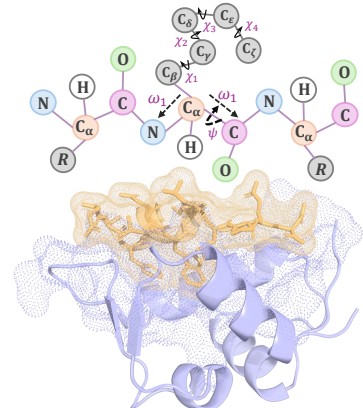

Figure 1: Top-down view of the receptor-peptide complex.

## 2 Preliminaries and Background

**Diffusion Models.** Let $q_0(\boldsymbol{x}_0)$ be a $d$-dimensional data distribution. The forward diffusion process [17, 40, 42] is defined by the following stochastic differential equation (SDE) with an initial condition $\boldsymbol{x}_0 \sim q_0$:

$$\mathrm{d}\boldsymbol{x}_t = f(t)\boldsymbol{x}_t\mathrm{d}t + g(t)\mathrm{d}\boldsymbol{\omega}_t, \tag{1}$$

where $t \in [0, T]$. $f(t)$ and $g(t)$ are scalar-valued drift and diffusion coefficients, respectively. $\boldsymbol{\omega}_t \in \mathbb{R}^d$ is a standard Wiener process. $q_0$ usually conforms to a random Gaussian noise. The

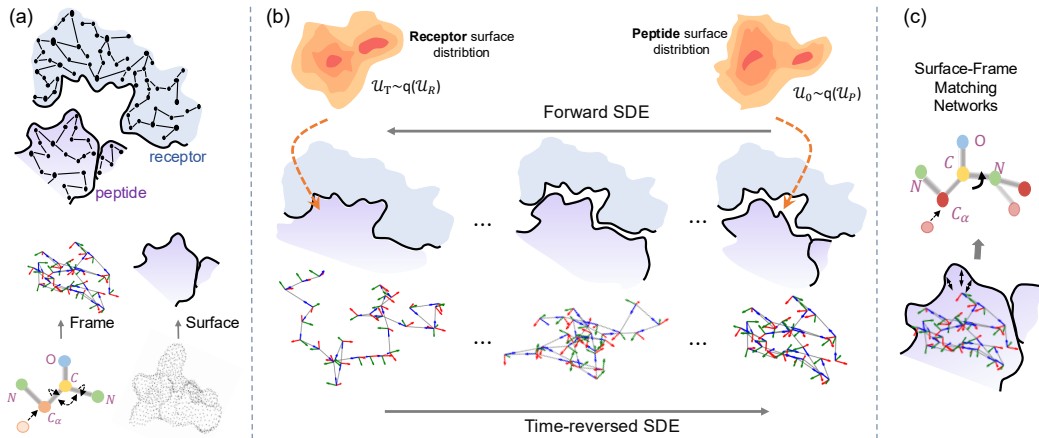

Figure 2: Illustration of the PepBridge architecture for joint surface-structure peptide generation. **(a)** The model processes receptor-ligand pairs through a top-down structure comprising molecular surface and frame components. **(b)** Two specialized diffusion models are employed simultaneously. A diffusion bridge model leverages the receptor surface as the starting point to generate peptide surfaces. An SE(3) diffusion model shoulders the responsibility of frame construction, which incorporates translation and torsion angles. **(c)** A surface-frame matching network facilitates the interaction between creased structures, while multi-modal diffusion reconstructs the complete peptide structure.

corresponding reverse-time SDE for sampling from $q_0(\boldsymbol{x}_0)$ is:

$$\mathrm{d}\boldsymbol{x}_t = [f(t)\boldsymbol{x}_t - g^2(t)\nabla_{\boldsymbol{x}_t}\log q_t(\boldsymbol{x}_t)]\mathrm{d}t + g(t)\mathrm{d}\hat{\boldsymbol{\omega}}_t, \tag{2}$$

where $\hat{\boldsymbol{\omega}}_t$ denotes the reverse-time Wiener process and $\nabla_{\boldsymbol{x}_t}\log q_t(\boldsymbol{x}_t)$ is the score function of the marginal density $q_t$.

**Diffusion Bridge Models.** Traditional diffusion models assume a Gaussian prior as the starting point for the generative process. However, in many practical scenarios, including protein design, the initial state may not follow a random Gaussian distribution, requiring a more flexible approach. To address this limitation, diffusion bridge models [7, 62, 63] provide a framework for modeling structured data distributions by matching the conditional score of a tractable bridge distribution. These models enable a transport between distributions through either a reverse SDE or a probability flow ordinary differential equation (ODE). For diffusion bridges with an initial condition $\boldsymbol{x}_0 \sim q_0 = p_{\mathrm{data}}$ and a terminal condition $\boldsymbol{x}_T = \boldsymbol{y}$, the forward process is:

$$\mathrm{d}\boldsymbol{x}_t = f(t)\boldsymbol{x}_t\mathrm{d}t + g^2(t)\nabla_{\boldsymbol{x}_t}\log q(\boldsymbol{x}_T = \boldsymbol{y}|\boldsymbol{x}_t) + g(t)\mathrm{d}\boldsymbol{\omega}_t, \tag{3}$$

where $\boldsymbol{y}$ is drawn from a prior distribution rather than Gaussian noise. The corresponding reverse SDE is:

$$\mathrm{d}\boldsymbol{x}_t = [f(t)\boldsymbol{x}_t - g^2(t)(\nabla_{\boldsymbol{x}_t}\log q(\boldsymbol{x}_t|\boldsymbol{x}_T = \boldsymbol{y}) - \nabla_{\boldsymbol{x}_t}\log q_{T|t}(\boldsymbol{x}_T = \boldsymbol{y}|\boldsymbol{x}_t)))]\mathrm{d}_t + g(t)\mathrm{d}\hat{\boldsymbol{\omega}}_t, \tag{4}$$

where $\nabla_{\boldsymbol{x}_t}\log q(\boldsymbol{x}_t|\boldsymbol{x}_T = \boldsymbol{y})$ represents the bridge score function and $\hat{\omega}$ is the reverse-time Wiener process.

## 3 Method

### 3.1 Problem Statement

The protein-peptide complex pair can be represented as $\mathcal{C} = \mathcal{P} \cup \mathcal{R}$, where $\mathcal{P}$ and $\mathcal{R}$ denote the peptide and receptor, respectively. For both the peptide and the receptor, we build the top-down (Upper-Bottom) structural representation $\mathcal{U} \cup \mathcal{B}$, where $\mathcal{U} = \{u_i\}_{N_\mathcal{U}}$ denotes the spatial surface point cloud and $\mathcal{B} = \{b_i\}_{N_\mathcal{B}}$ denotes the residue-level structure. The bottom structure $\mathcal{B}$ is composed of amino acid residues, where each residue $b_i$ is characterized by its backbone frame, residue type, and side-chain dihedral angles [12], as illustrated in Figure 1. The goal of target-aware peptide generation is to learn a probabilistic model that captures the distribution over peptide top-down structures, $p(\mathcal{P}|\mathcal{R})$, conditioned on the receptor as a structural and biochemical reference.

## 3.2 Surface Diffusion Bridge

Peptides typically fold into complementary shapes when binding to their receptors [29]. The geometric and biochemical properties of the receptor's binding site impose natural constraints on the conformational space of compatible peptide structures. For the sake of leveraging this relationship, we develop a tailored diffusion bridge model [63] that treats the receptor surface $\mathcal{U}_{\mathcal{R}}$ as a prior distribution to generate energetically favorable peptide conformations with complementary binding surfaces $\mathcal{U}_{\mathcal{P}}$.

**Surface Representation.** Firstly, we devise a pipeline for molecular surface processing and point cloud extraction. Starting with a protein structure in PDB format, we use PyMol [10] to generate solvent-accessible surface representations. The probe molecular surface approximates both the Solvent-Accessible Surface Area (SASA) and Solvent Excluded Surface (SES). The resulting point cloud consists of surface points $\boldsymbol{u}_i$, each annotated with 3D spatial coordinates and physicochemical features, including hydrophobicity and hydrogen bonding potential [13, 31].

**Diffusion Bridge via $h$-transform.** The surfaces of receptor and peptide exhibit close interactions, with their distributions $p_{\mathcal{P}}$ and $p_{\mathcal{R}}$ naturally forming pairs. We model their surface fit by reconstructing a stochastic trajectory between observed positions. Specifically, let $(\mathcal{U}_0, \mathcal{U}_T)$ denote a pair of surface datasets with empirical marginal distributions $p_0$ and $p_T$ at times $t = 0$ and $t = T$, respectively. Given these endpoint distributions, our objective is to reconstruct the continuous-time stochastic process $p_t$ over $t \in [0, T]$ that interpolates between $p_0$ and $p_T$. Using Doob's h-transform [11, 38], we define a surface diffusion process that transitions from the peptide surface $\mathcal{U}_0$ to the receptor surface $\mathcal{U}_T$. The forward SDE is given by:

$$d\mathcal{U}_t = f(\mathcal{U}_t, t)dt + g(t)^2 \boldsymbol{h}(\mathcal{U}_t, t, \mathcal{U}_{\mathcal{R}}, T) + g(t)d\boldsymbol{w}_t, \tag{5}$$

where $\boldsymbol{h}(\mathcal{U}_t, t, \mathcal{U}_{\mathcal{R}}, T) = \nabla_{\mathcal{U}_t} \log p(\mathcal{U}_T | \mathcal{U}_t)|_{\mathcal{U}_t = \mathcal{U}_{\mathcal{P}}, \mathcal{U}_T = \mathcal{U}_{\mathcal{R}}}$ represents the gradient of the logarithmic transition kernel. The corresponding time-reversed SDE is constructed as

$$d\mathcal{U}_t = [f(\mathcal{U}_t, t) + g(t)^2(\boldsymbol{s}(\mathcal{U}_t, t, \mathcal{U}_{\mathcal{R}}, T) - \boldsymbol{h}(\mathcal{U}_t, t, \mathcal{U}_{\mathcal{R}}, T))]dt + g(t)d\hat{\boldsymbol{w}}_t. \tag{6}$$

As shown in Figure 2, we use the receptor surface as an informative prior in place of Gaussian noise, enabling more efficient generation of complementary peptide surfaces tailored to the binding site. Accordingly, we define the forward transition kernel as $q(\mathcal{U}_t | \mathcal{U}_0, \mathcal{U}_T) = \mathcal{N}(\hat{\mu}_t, \hat{\sigma}_t^2 \boldsymbol{I})$, where $\hat{\mu}_t = \frac{\alpha_t}{\alpha_T} \frac{\mathrm{SNR}_T}{\mathrm{SNR}_t} \mathcal{U}_T + \alpha_t \mathcal{U}_0 (1 - \frac{\mathrm{SNR}_T}{\mathrm{SNR}_t})$ and $\hat{\sigma}_t^2 = \sigma_t^2 (1 - \frac{\mathrm{SNR}_T}{\mathrm{SNR}_t})$. $\alpha_t$ is a fixed signal scaling factor and normally takes the value of 1.0. $\sigma_t$ defines the noise schedule, and $\mathrm{SNR}_t = \alpha_t^2 / \sigma_t^2$ denotes the signal-to-noise ratio at time $t$. During the sampling process, we start from $p_T \sim p_{\mathcal{U}_{\mathcal{R}}}$ and approximate the score via $\boldsymbol{s}(\mathcal{U}_t, t, \mathcal{U}_{\mathcal{R}}, T) = \nabla_{\mathcal{U}_t} \log q(\mathcal{U}_t | \mathcal{U}_T)|_{\mathcal{U}_t = \mathcal{U}_{\mathcal{P}}, \mathcal{U}_T = \mathcal{U}_{\mathcal{R}}}$, where $q(\mathcal{U}_t | \mathcal{U}_T) = \int_{\mathcal{U}_0} q(\mathcal{U}_t | \mathcal{U}_0, \mathcal{U}_T) q_{\mathrm{data}}(\mathcal{U}_0 | \mathcal{U}_T) d\mathcal{U}_0$.

**Surface Generation Loss.** We employ denoising score-matching [42] with neural network approximation of the true score $\nabla_{\mathcal{U}_t} \log q(\cdot)$, leading to the surface generation loss $\mathcal{L}_{\mathcal{U}}$ as:

$$\mathcal{L}_{\mathcal{U}} = \mathbb{E}_t \mathbb{E}_{\mathcal{U}_0, \mathcal{U}_R \sim p_{\mathrm{data}}(\mathcal{U}_0, \mathcal{U}_R)} \mathbb{E}_{\mathcal{U}_t \sim q(\mathcal{U}_t | \mathcal{U}_0, \mathcal{U}_T = \mathcal{U}_R)} [w(t) \parallel \boldsymbol{s}_\theta(\mathcal{U}_t, \mathcal{U}_T, t) - \nabla_{\mathcal{U}_t} \log q(\mathcal{U}_t | \mathcal{U}_0, \mathcal{U}_T) \parallel^2],$$

where $q(\mathcal{U}_t | \mathcal{U}_0, \mathcal{U}_T)$ is the previously defined forward transition kernel and $w(t)$ is the time-dependent weighting function. $\boldsymbol{s}_\theta(\cdot)$ represents the parameterized geometric network, with detailed information provided in Appendix D.

## 3.3 Bottom Structure Diffusion Generation

**Bottom Structure Parameterization.** Following AlphaFold2 and recent works [26, 57, 58], we parameterize the peptide backbone using four key atoms $\mathrm{N}^\star, \mathrm{C}_\alpha^\star, \mathrm{C}^\star, \mathrm{O}^\star$, which form a rigid body frame. The rigid frame centered at $\mathrm{C}_\alpha$ atom, i.e., $\mathrm{C}_\alpha^\star = (0, 0, 0)$. Applying an SE(3) transformation $\mathcal{T}_n$ to the local backbone frame of residue $n$ yields the global atomic coordinates:

$$[\mathrm{N}_n, \mathrm{C}_n, (\mathrm{C}_\alpha)_n] = \mathcal{T}_n \cdot [\mathrm{N}^\star, \mathrm{C}^\star, \mathrm{C}_\alpha^\star], \quad \mathrm{O}_n = \mathcal{T}_n \cdot \mathcal{T}_{\mathrm{psi}}^\star(\psi_n) \cdot \mathrm{O}^\star,$$

where $\mathcal{T}_n = (r_n, m_n)$ consists of a rotation matrix $r_n \in \mathrm{SO}(3)$ and a translation vector $m_n \in \mathbb{R}^3$. The transformation $\mathcal{T}_{\mathrm{psi}}^\star(\psi_n) = (r(\psi_n), m_{psi})$ encodes a rotation of $\mathrm{O}^\star$ around the $\mathrm{C}_\alpha - \mathrm{C}$ bond by torsion angle $\psi_n$. The amino acid type of the $i$-th residue $a_i \in \{1..20\}$ is determined by the side-chain R group. The side-chain conformation is governed by torsion angles between side-chain

atoms, represented as $\chi \in [0, 2\pi)^4$. A detailed description is provided in Appendices A and B. Our approach to bottom structure prediction integrates three components: a multi-modal diffusion process on SE(3) for backbone prediction, a torus diffusion model for torsion angle generation, and a logit-normal diffusion model for residue identity prediction.

**Frame Structure Generation.** Given a sequence of $N$ rigid transformations $\mathcal{T} = [\mathcal{T}_1, ..., \mathcal{T}_N] \in \mathrm{SE}(3)^N$, we model their distribution using Riemannian diffusion on $\mathrm{SE}(3)^N$ [8]. The forward diffusion process on the SE(3)-invariant measure is:

$$\mathrm{d}\mathcal{T}_t = [0, -P\frac{1}{2}\boldsymbol{m}_t]\mathrm{d}t + [\mathrm{d}\boldsymbol{B}_t^{\mathrm{SO}(3)}, \mathrm{d}\boldsymbol{B}_t^{\mathbb{R}^3}], \tag{7}$$

where $\boldsymbol{B}_t^{\mathrm{SE}(3)} = [\boldsymbol{B}_t^{\mathrm{SO}(3)}, \boldsymbol{B}_t^{\mathbb{R}^3}]$ represent Brownian motion on SO(3) and $\mathbb{R}^3$, and $P \in \mathbb{R}^{3N \times 3N}$ is a projection matrix removing the center of mass $\frac{1}{N}\sum_{n=1}^N m_n$. In Appendix C, we show the choice of metric on SE(3), which allows decomposing the process into independent translational and rotational components. The backward process is given by :

$$\nabla \boldsymbol{r}_t = \nabla_{\boldsymbol{r}} \log p_{T_F-t}(\mathcal{T}_t)\mathrm{d}t + \mathrm{d}\boldsymbol{B}_t^{\mathrm{SO}(3)}, \nabla \boldsymbol{m}_t = P\{\frac{1}{2}\boldsymbol{m}_t + \nabla_{\boldsymbol{m}} \log p_{T_F-t}(\mathcal{T}_t)\}\mathrm{d}t + P\mathrm{d}\boldsymbol{B}_t^{\mathrm{SO}(3)},$$

where $T_F$ denotes the final time step. We show more details about the training and sampling in Appendix C.

**Structure Generation Loss.** Backbone generation is supervised by a denoising score matching loss $\mathcal{L}_{\mathcal{T}}$, combining rotation and translation components. The loss function for the rotation component is expressed mathematically as:

$$\mathcal{L}_{\boldsymbol{r}}(\theta) = \mathbb{E}_t \mathbb{E}_{\boldsymbol{r}_0,\boldsymbol{r}_T \sim p_{\mathrm{data}}(\boldsymbol{r}_0,\boldsymbol{r}_T)} \mathbb{E}_{\boldsymbol{r}_t \sim p_{\mathrm{data}}(r_t|\boldsymbol{r}_0,\boldsymbol{r}_T)}[\lambda_t^r \parallel \nabla \log p_{t|0}(\boldsymbol{r}_t|\boldsymbol{r}_0) - s_\theta(t, \boldsymbol{r}_t) \parallel^2], \tag{8}$$

where the rotation weighting schedule is formulated as $\lambda_t^r = 1/E[\parallel \nabla \log p_{t|0}(\boldsymbol{r}_t, \boldsymbol{r}_0) \parallel_{\mathrm{SO}(3)}^2]$. Meanwhile, the translation loss component is written as:

$$\mathcal{L}_{\boldsymbol{m}}(\theta) = \mathbb{E}_t \mathbb{E}_{\boldsymbol{m}_0,\boldsymbol{m}_T} \mathbb{E}_{\boldsymbol{m}_t}[\parallel \boldsymbol{m}_0 - s_\theta(t, \boldsymbol{m}_t) \parallel^2]. \tag{9}$$

We observe that directly regressing $C_\alpha$ coordinates improves stability over using score matching for $\boldsymbol{m}_t$. This operation ensures that the generated backbone structures remain physically consistent with the receptor-ligand complex. The translation weight schedule is defined as $\lambda_t^m = (1 - e^{-t}/e^{-t/2})$.

**Residue Type Prediction.** Residue types $a^j$ are modeled as categorical variables embedded in logit space via a sharp one-hot encoding: $\boldsymbol{v}^j \in \mathbb{R}^{20}$, where $\mathbf{v}^j[i] = K$ if $i = a^j$, otherwise $-K$, with $K > 0$ a fixed constant. Applying a softmax transformation to $\boldsymbol{v}^j$ yields a distribution over the 20-simplex, sharply peaked at the index corresponding to $a^j$. This effectively embeds the discrete residue type as a concentrated categorical distribution on the simplex. We define a forward diffusion process in logit space: $q(\mathbf{v}_t^j|\mathbf{v}_{t-1}^j) = \mathcal{N}(\mathbf{v}_t^j; \sqrt{\alpha_t}\mathbf{v}_{t-1}^j, (1-\alpha_t)K^2I)$, with the prior $p(\boldsymbol{v}_T^j) = \mathcal{N}(0, K^2I)$, corresponding to a logit-normal distribution [3]. The reverse process recover the categorical residue types by sampling from the softmax output:

$$\mathbf{v}_{t-1}^j = \sqrt{\bar{\alpha}_t}\boldsymbol{v}_0^j + \sqrt{1 - \bar{\alpha}_t}\boldsymbol{\epsilon}, \quad \boldsymbol{\epsilon} \sim \mathcal{N}(0, K^2I), \tag{10}$$

with final prediction $a^j \sim \mathrm{softmax}(\mathbf{v}_0^j)$. The model is trained to predict $\boldsymbol{\epsilon}$ using the following loss function:

$$\mathcal{L}_{\mathrm{type}}^j = \mathbb{E}_{t,\mathbf{v}_0^j,\boldsymbol{\epsilon}} \parallel \boldsymbol{\epsilon}_\theta^{\mathrm{type}}(\mathbf{v}_t^j, t) - \boldsymbol{\epsilon} \parallel_2^2, \tag{11}$$

**Torsion Angle Prediction.** The torsion vector $\boldsymbol{\chi}_i \in [0, 2\pi)^5$ consists of four side-chain angles and one backbone torsion angle $\psi$. To model angular diffusion, we apply the DDPM framework on the torus, using a wrap function to maintain values within $[-\pi, \pi)$: $\mathrm{wrap}(\boldsymbol{\chi}) = (\boldsymbol{\chi} + \pi)\%(2\pi) - \pi$. The forward process perturbs the angles with Gaussian noise: $\boldsymbol{\chi}_t = \mathrm{wrap}(\sqrt{\bar{\alpha}_t}\boldsymbol{\chi}_{t-1} + \sqrt{1 - \bar{\alpha}_t}\boldsymbol{\epsilon}), \boldsymbol{\epsilon} \sim \mathcal{N}(0, I)$, where $\bar{\alpha}_t = \prod_{s=1}^t (1 - \beta_s)$ and $\beta_t \in [0, 1]$ is the noise schedule. The reverse process approximates the posterior distribution: $p(\boldsymbol{\chi}^{t-1}|\boldsymbol{\chi}^t) = \mathrm{wrapnormal}[\mu_\theta(\boldsymbol{\chi}^t, t), \sigma_t^2 I]$, where $\mu_\theta(\boldsymbol{\chi}^t, t) = \mathrm{wrap}\left[\frac{1}{\sqrt{\alpha_t}}(\boldsymbol{\chi}_t - \frac{1-\alpha_t}{\sqrt{1-\bar{\alpha}_t}}\boldsymbol{\epsilon}_\theta^{\mathrm{ang}}(\chi_t, t))\right]$ is the model-predicted mean. The network predicts the noise vector $\boldsymbol{\epsilon}_\theta$ and the training loss is defined as:

$$\mathcal{L}_{\mathrm{ang}} = \mathbb{E}_{t,\boldsymbol{\chi}_0,\boldsymbol{\epsilon}_t} \parallel \mathrm{wrap}(\boldsymbol{\epsilon}_\theta^{\mathrm{ang}}(\boldsymbol{\chi}_t, t) - \boldsymbol{\epsilon}) \parallel^2 . \tag{12}$$

### 3.4 Shape-Frame Matching Network

Our co-design network implements iterative updates to the top-down structure through a Shape-Frame Matching Network. We denote its $l$-th layer as $\mathrm{SFMNet}(\{h_{\mathcal{U}}^{(l)}, x_{\mathcal{U}}^{(l)}\}, \{h_{\mathcal{B}}^{(l)}, x_{\mathcal{B}}^{(l)}\})$. Here, $x_{\mathcal{U}}^{(l)} \in \mathbb{R}^{N_{\mathcal{U}} \times 3}$, $x_{\mathcal{B}}^{(l)} \in \mathbb{R}^{N_{\mathcal{B}} \times 3}$ are transformed coordinates of surface and frame, while $h_{\mathcal{U}}^{(l)} \in \mathbb{R}^{N_{\mathcal{U}} \times d_{\mathcal{U}}}$ and $h_{\mathcal{B}}^{(l)} \in \mathbb{R}^{N_{\mathcal{B}} \times d_{\mathcal{B}}}$ are feature embeddings. This architecture jointly transforms both features and 3D coordinates to perform interaction between surface points and backbone frames. By stacking $L$ layers of SFMNet, we ensure equivariant updates to the protein's top-down structure. The single $l$-th layer is formulated as a variant of the 3D equivariant graph neural network (EGNN) [39, 54]. First of all, we get the attention score $\mathbf{att}_{ij}^{(l)} = \frac{1}{\sqrt{d_{\mathcal{U}}}}(h_{b_i}^{(l)} W_Q)(h_{u_j}^{(l)} W_K + g_{ij})$, where $W_Q \in \mathbb{R}^{d_{\mathcal{B}} \times d_{\mathcal{U}}}$ and $W_K \in \mathbb{R}^{d_{\mathcal{U}} \times d_{\mathcal{U}}}$ are learnable matrices. A geometric structural embedding $g_{ij} \in \mathcal{R}^{d_{\mathcal{U}}}$ is incorporated into the attention computation. It is obtained by feeding the geodesic distance into a multi-layer perceptron (MLP) as $g_{ij} = \mathrm{MLP}(\| x_{b_i} - x_{u_j} \|)$. Then we aggregate the message from both backbone frames and surface points, and update the backbone node features $h_{\mathcal{B}}^{(l)}$ as

$$\nu_{b_i,u_j} = \phi_\nu(h_{b_i}^{(l)}, h_{u_j}^{(l)}, \mathbf{att}_{ij}^{(l)}, t, \| x_{b_i} - x_{u_j} \|), \quad h_{b_i}^{(l+1)} = \phi_h(h_{b_i}^{(l)}, \sum_{j \in \mathcal{N}(b_i)} \nu_{b_i,u_j}),$$

where $\phi_\nu(\cdot)$ and $\phi_h(\cdot)$ are two additional MLPs to accumulate the adjacent messages and features. $\mathcal{N}(b_i)$ is the neighborhood set of frame node $b_i$ that contains all surface points $\{u_j| \| x_{b_i} - x_{u_j} \| \leq \gamma\}$ that interact with this residue, where $\gamma$ is the distance threshold. After that, we calculate the shift of translation $\Delta m^l$ and rotation $\Delta r^l$, adding them to the original values:

$$m_{b_i}^{(l+1)} = m_{b_i}^{(l)} + \phi_m(h_{b_i}^{(l+1)}), \quad r_{b_i}^{(l+1)} = r_{b_i}^{(l)} + \phi_r(h_{b_i}^{(l+1)}). \tag{13}$$

### 3.5 Overall Training Loss

The complete loss function combines the surface and bottom structure loss functions:

$$L_{\mathrm{total}} = \mu^T[L_{\mathcal{U}}, L_r, L_m, L_{\mathrm{type}}, L_{\mathrm{ang}}], \tag{14}$$

where $\mu$ denotes the set of weighting hyperparameters balancing each loss term. This formulation enables joint optimization over both surface geometry and residue structure for comprehensive protein structure prediction.

## 4 Experiments

This section presents comprehensive experimental evaluations to demonstrate the efficacy of our proposed method. Our investigation addresses three fundamental questions: **Q1**: How do the generated samples perform in terms of quality? **Q2**: Does the method generate physically valid samples? **Q3**: What is the impact of key architectural decisions on model performance? We evaluate PepBridge and baseline methods on two tasks: (1) *Surface-Structure Joint-design.* Generation of peptide structure and surface conditional on a specified receptor binding site. This task involves the simultaneous generation of peptide surface characteristics and structural conformations, conditional on a specified receptor binding site. (2) *Side-chain Packing.* Prediction of optimal side-chain angles for peptides when they are bound to a specified receptor site.

### 4.1 Experimental Setup

**Dataset.** The evaluation utilized the PepMerge dataset [25], a collection derived from the integration of PepBDB [52] and Q-BioLip [51] databases. Following the protocol established in [25], we implemented rigorous filtering criteria: eliminating redundant entries, excluding structural data with resolution exceeding 4 Å, and selecting peptides with sequence lengths between 3 and 25 residues. We evaluate the generation model's performance using multiple criteria to ensure candidates exhibit high diversity and novelty while maintaining validity and desired distributional properties. For each receptor, we generate 40 candidates. Additional dataset details are provided in Appendix E.1.

**Baselines.** We compare our approach against two distinct categories of state-of-the-art protein design methods. The first category consists of backbone-centric approaches that do not explicitly model

Table 1: Evaluation on surface-structure joint-design. On each target, 40 candidates are generated for evaluation. Div., Aff. and Stab. are abbreviations for diversity, affinity and stability, respectively.

| | $\text{Div}_{\text{stru}}$ (↑) | Aff. % (↑) | Stab. % (↑) | RMSD Å(↓) | BSR (↑) |
|---|---|---|---|---|---|
| ProteinGenerator | 0.54 | 13.47 | 23.48 | 4.35 | 24.62 |
| RFDiffusion | 0.38 | 16.53 | 26.82 | 4.17 | 26.71 |
| Chroma (RIA) | 0.59 | 17.96 | 16.69 | 3.97 | 74.12 |
| PPFLOW | 0.53 | 17.62 | 17.25 | 2.94 | 78.72 |
| PepGLAD | 0.32 | 10.47 | 20.39 | 3.83 | 19.34 |
| PepFlow w/Bb | 0.64 | 18.10 | 14.04 | 2.30 | 82.17 |
| PepFlow w/Bb+Seq | 0.50 | 19.39 | 19.20 | 2.21 | 85.19 |
| PepFlow w/Bb+Seq+Ang | 0.42 | 21.37 | 18.15 | 2.07 | 86.89 |
| PepBridge w/Bb+surf | 0.60 | 20.07 | 21.75 | 2.04 | 84.62 |
| PepBridge w/Bb+Seq+surf | 0.62 | 22.07 | 20.71 | 2.18 | 84.91 |
| PepBridge w/Bb+Seq+Ang+surf | 0.59 | 19.16 | 25.02 | 2.19 | 83.90 |

side-chain conformations, such as RFDiffusion [50], ProteinGenerator [27] and PPFLOW [26]. RFDiffusion generates backbones via diffusion, followed by sequence prediction with ProteinMPNN [6], while ProteinGenerator jointly models sequence and backbone. PPFLOW [26] conditions on the target receptor and uses conditional flow matching on torus manifolds to generate peptide backbones by modeling torsional geometry. The second category comprises full-atom models like PepGLAD [21], Chroma [19], and PepFlow [25], which explicitly generate side-chain conformations. To evaluate Chroma, we used its conditional generation with receptor interaction area (RIA) as the conditioning input. Further details are provided in Appendix E.2.

## 4.2 Surface-Structure Joint-Design

**Metrics.** We evaluate the validity of the generated backbone structures using the following metrics. (1) *Diversity*: The average of one minus the pairwise TM-score [59] between generated peptides. A higher diversity score indicates higher dissimilarity and greater structural variety among the generated peptides. (2) *Affinity*: Binding affinity is evaluated using Rosetta's energy function [2], measured in kcal/mol. We report that the proportion of generated peptides with binding energies is lower than that of the native peptide. (3) *Stability*: The percentage of generated peptide-protein complexes with total energy lower than the native complex. (4) $\text{RMSD}_{C_\alpha}$: The root mean square deviation of $C_\alpha$ atom coordinates between the generated and reference peptide structures, measured in Ångströms (Å). (5) *BSR*: The binding site ratio, which measures the overlap between the binding sites of the generated and native peptides on the target protein.

**Results.** As shown in Table 1, PepBridge consistently benefits from explicit surface conditioning. Among our variants, PepBridge w/Bb+Seq+surf attains the highest affinity, and PepBridge w/Bb+surf yields the lowest RMSD across all methods, indicating strong geometric fidelity to native-like docked poses. PepBridge w/Bb+Seq+Ang+surf ranks second in stability at 25.02% and maintains competitive structural diversity. Surface-aware generation consistently outperforms backbone-only counterparts. Relative to PepFlow w/Bb, PepBridge w/Bb+surf increases affinity and markedly reduces RMSD. PepBridge w/Bb+Seq+surf continues this trend, further reducing RMSD while raising affinity. PepBridge variants deliver strong RMSD, affinity, and stability, translating into competitive enrichment (BSR). Modeling interface geometry through joint surface–backbone conditioning proves especially effective for enhancing conformational accuracy and binding complementarity.

## 4.3 Ablation Study

To comprehensively assess the contribution of each component in our proposed model, we conduct an ablation study by systematically removing or modifying key elements. (1) *-bridge/+vanilla diffusion*: The denoising diffusion bridge model is replaced with a standard diffusion model, which diminishes the incorporation of receptor surface geometry as prior information. (2) *-bridge/+CFG diffusion*: The denoising diffusion bridge model is replaced with a classifier-free guidance diffusion model [16], which enables receptor-guided conditional generation. However, this approach still maintains a Gaussian prior distribution, rather than incorporating explicit geometric priors. (3)

Table 2: Ablations on different components of PepBridge, where the best model is in **bold**.

| Ablations | $\text{Div}_{\text{stru}}$ (↑) | $\text{Div}_{\text{surf}}$ (↑) | Aff. % (↑) | Stab. % (↑) | RMSD Å(↓) | BSR (↑) | Con. (↑) |
|---|---|---|---|---|---|---|---|
| PepBridge | **0.59** | **0.46** | **19.16** | **25.02** | **2.19** | **83.90** | **0.43** |
| -bridge/+vanilla diffusion | 0.42 | 0.39 | 15.97 | 17.72 | 3.18 | 46.21 | 0.31 |
| -bridge/+CFG diffusion | 0.39 | 0.41 | 16.38 | 19.32 | 3.46 | 57.39 | 0.36 |
| -surface context | 0.51 | – | 16.17 | 15.37 | 4.21 | 31.37 | – |
| -surface&frame matching | 0.42 | 0.35 | 14.82 | 22.41 | 3.71 | 54.71 | 0.25 |

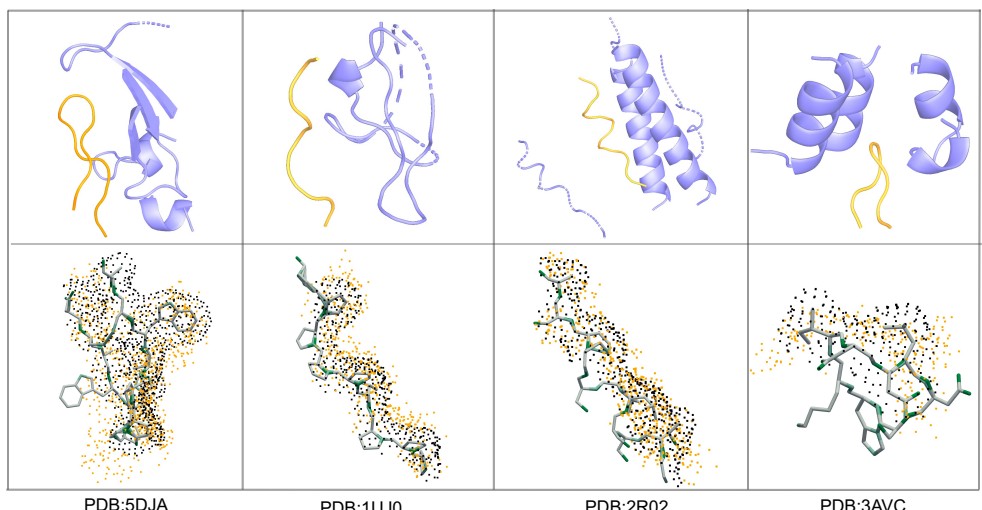

PDB:5DJA          PDB:1UJ0          PDB:2R02          PDB:3AVC

Figure 3: Visualization of generated peptides by our PepBridge. **Top:** Generated peptides (in orange) for receptors (in purple). The PDB ID of the receptors are 5DJA, 1UJ0, 2R02, and 3AVC. **Bottom:** The generated backbone structure and surface. The ground-truth surface structure (in black) and generated surface (in orange) are shown to compare the ability of the interface caption.

*-surface context*: Surface context information is removed, leaving only the backbone generation component. (4) *-surface&frame matching*: The surface-frame matching mechanism is excluded during the training of the denoising model. We also evaluate additional metrics to analyze model performance: (1) *Consistency*: Surface-structure consistency is quantified using Cramér [5], which measures the association between surface and structure clustering labels. Higher consistency values indicate that candidates with similar structures tend to have similar surfaces, suggesting that the model effectively captures the interdependence between backbone structures and surface features. (2) *Surface Diversity*: $\text{Div}_{\text{surf}}$ is computed based on surface alignment similarities. Detailed information about the metrics can be found in Appendix E.3.

Table 2 records the ablation results. It shows that during surface generation, the basic diffusion model struggles to capture the correct distribution compared to the diffusion bridge model, resulting in instability and low consistency in the generated structures. Replacing the denoising diffusion bridge model with the classifier-free guidance diffusion model still fails to capture the surface distribution accurately. When eliminating the surface context and only generating the backbone, the performance on affinity, RMSD, and BSR drops dramatically, since it can not get enough binding set information. Without surface guidance, the model will generate unrealistic and unstable peptides. As for the component of the surface-frame matching mechanism, through the result, we can know that the network helps to achieve high stability and enhance consistency between the surface and backbone. It also helps to reduce $\text{RMSD}_{C_\alpha}$ by providing effective structural patterns.

## 4.4 Visualization

We further present four examples of generated peptides in Figure 4.3. We observe that PepBridge produces peptides with proper geometries and positions. The generated surface exhibits similar

Table 3: Evaluation of different methods in the side-chain packing task, where the best and the suboptimal approaches are **bolded** and underlined, respectively.

| | Angle MAE °($\downarrow$) | | | | Angle Accuracy %($\uparrow$) | | |
|---|---|---|---|---|---|---|---|
| | $\chi_1$ | $\chi_2$ | $\chi_3$ | $\chi_4$ | All residues | Core residues | Surface residues |
| SCWRL4 | 29.79 | 30.12 | 52.38 | 62.03 | 45.93 | 66.25 | 34.59 |
| DLPacker | 28.35 | 32.62 | 54.69 | 59.60 | 49.00 | 68.03 | 39.56 |
| AttnPacker | 29.61 | 28.83 | 47.66 | 53.64 | 47.53 | 71.65 | 38.90 |
| DiffPack | 26.29 | 29.57 | 47.64 | 56.85 | 55.86 | **79.62** | 41.31 |
| PepFlow w/Bb+Seq+Ang | 27.61 | **25.60** | 48.20 | 54.02 | 54.29 | 70.47 | 44.06 |
| PepBridge (ours) | **25.96** | 26.76 | **46.81** | **52.95** | **56.71** | 73.79 | **46.17** |

conformation with the ground-truth surface, which show the ability to interact with the target binding sites and capture the right shape. We provide additional experiments in Appendix F.

## 4.5 Side-chain Packing

The backbone prior state is initialized using a native peptide structure and subsequently generate the surface and side-chain angles. We compare our approach against established methods including SCWRL4 [23], DLPacker [34], AttnPacker [32], DiffPack [60], and PepFlow [25]. The evaluation metrics include: (1) *Angle MAE*, which quantifies the mean absolute error between predicted and ground-truth angles, and (2) *Angle Accuracy*, which considers torsion angles correct when their deviation falls within $20°$. Following the methodology of [60], we present results for core residues, surface residues, and all residues. For each peptide, we generate an ensemble of 64 conformations.

Table 3 presents the comparative results. Our model demonstrates superior performance in predicting $\chi_1$, $\chi_3$, and $\chi_4$ angles. Particularly, PepBridge reduces the prediction error by 2% to 52.95 for the most challenging angle $\chi_4$. This suggests that the integration of surface-level information enhances side-chain angle prediction accuracy. Furthermore, the model exhibits particularly strong performance in surface residue prediction with a high accuracy of 46.17%, indicating that PepBridge effectively captures spatial relationships in interface regions. It also attains the best overall accuracy of 56.71%, which significantly improves the prior state-of-the-art PepFlow by 4.45%.

## 5 Related Works

**Computational Protein Design** Sequence-based and structure-based approaches are two main trajectories in computational protein design. The former models amino acid chains by language models [15, 30, 49], whereas the latter models the 3D geometry. Notable structure-based methods include FoldingDiff [55], which represents protein backbones through sequential angles, and RFD-iffusion [50], which employs varied diffusion schemes for backbone generation. FrameDiff [58] advanced this field by developing SE(3)-invariant diffusion models for protein modeling. Flow models have also shown promise in backbone design, as demonstrated by FOLDFLOW [4] and PPFLOW [26]. Recently, sequence-structure co-design has gained attention, with models such as PepGLAD [21] and PepFlow [25]. PepHAR [24] further targets peptide binders via hotspot sampling and multifragment autoregressive extension, enforcing geometric validity. Surface-conditioned protein modeling represents the latest frontier in this field. Surface-VQMAE [53] introduced a Transformer-based architecture that integrates surface geometry and captures patch-level relations. Despite those achievements, the joint design of the surface and structure remains an unexplored area, and we position our study as a pioneering effort in this direction.

**Surface Context in design ligands** Classical methods explicitly model geometric and physicochemical complementarity (e.g., shape matching, lock-and-key). Mesh-based learning like MaSIF [13] encodes surfaces with handcrafted geometric/chemical descriptors, while dMaSIF [47] accelerates this via point-cloud surfaces with atom-level features. Recent generative methods like ShEPhERD [1] and DSR [46] incorporate surface features into diffusion frameworks, and SurfPro [44] generates sequences conditioned on known surfaces. However, these approaches typically assume strict complementarity, require hand-crafted features, or depend on ground-truth surfaces. PepBridge instead uses denoising diffusion bridge models to learn data-driven mappings between receptor and ligand

interfaces, capturing flexible, non-complementary interactions crucial for peptide binding where induced fit dominates.

**Diffusion Models** Diffusion models have emerged as powerful probabilistic generative models [17, 41–43]. Recent advances have extended these models to handle data with inherent invariances [37, 56], as well as to discrete domains [28, 33]. In parallel, manifold-aware diffusion models have been introduced, including the Riemannian Score-Based Generative Model (RSGM)[8] and the Riemannian Diffusion Model (RDM)[18]. A significant variant, diffusion bridge models [7, 62, 63], enables interpolation between paired endpoint distributions. In this work, we employ a diffusion bridge model to construct a stochastic bridge between receptor and ligand distributions, enabling the generation of complementary, high-quality samples.

## 6 Conclusion

This work presents PepBridge, a novel framework conditioned on receptor structures for joint protein surface-backbone design. We employ a stochastic bridge process between receptor and ligand surfaces with tractable marginal distributions, where the model learns by matching conditional scores of the bridge distribution. For backbone generation, an SE(3) diffusion model is used to predict frame geometry. The Surface-Frame Matching Network enables bidirectional information flow between surface and backbone levels, facilitating coherent structural development. The superior performance of PepBridge highlighs the advantages of shape-driven generation in target protein design.

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

# A Backbone Representation

As introduced in 3.3 section, every frame is composed by four atomic group $N^\star, C^\star_\alpha, C^\star, O^\star$, which is idealized atom coordinates that assumes chemically idealized bond angles and lengths.

We use the tuple $T = (r, m)$ to denote the Euclidean transformations corresponding to frames, where $r \in \mathrm{SO}(3)$ for the rotation and $m \in \mathbb{R}^3$ for the translation components. We use the dot product operator ($\cdot$) to denote application of a transformation to the position of frame $b \in \mathbb{R}^3$:

$$\begin{aligned} \hat{b} &= T \cdot b \\ &= (r, m) \cdot b \\ &= rb + m. \end{aligned}$$

The composition of Euclidean transformations denoted as:

$$\begin{aligned} T &= T_1 \cdot T_2 \\ (r, m) &= (r_1, m_1) \cdot (r_2, m_2) \\ &= (r_1 r_2, \ r_1 m_2 + m_1). \end{aligned}$$

The group inverse of the transform $T$ is denoted as:

$$\begin{aligned} T^{-1} &= (r, m)^{-1} \\ &= (r^{-1}, \ -r^{-1} m) \end{aligned}$$

The tuple transforms a position in local coordinates $b_{\text{local}} \in \mathbb{R}^3$ to a position in global coordinates $b_{\text{global}} \in \mathbb{R}^3$ as

$$\begin{aligned} b_{\text{global}} &= T \cdot b_{\text{local}} \\ &= r b_{\text{local}} + m \,. \end{aligned}$$

In local position of frame, the bond angles and lengths values differ slightly per amino acid type. Follow [58] and [57], we set the local coordinates as:

$$\begin{aligned} N^\star &= (-0.525, 1.363, 0.0) \\ C^\star_\alpha &= (0.0, 0.0, 0.0) \\ C^\star &= (1.526, 0.0, 0.0) \\ O^\star &= (0.627, 1.062, 0.0) \end{aligned} \tag{15}$$

where $C^\star_\alpha$ is central in protein backbones, connecting $N^\star$ and $C^\star$ groups. Using the transformation $T_n$, we manipulate idealized coordinates to construct global coordinates of backbone atoms for residue $n$ via:

$$\begin{aligned} [N_n, C_n, (C_\alpha)_n, O_n] = \big[ &T_n \cdot N^\star, \ T_n \cdot C^\star, \\ &T_n \cdot C^\star_\alpha, \ T_n \cdot T^\star_{\text{psi}}(\psi_n) \cdot O^\star \big]. \end{aligned} \tag{16}$$

Given the coordinates of three atoms $[N_n, C_n, (C_\alpha)_n]$, we construct a local rigid frame using a Gram-Schmidt process:

$$\begin{aligned} \omega_1 &= C_n - (C_\alpha)_n, & \omega_2 &= N_n - (C_\alpha)_n \\ e_1 &= \omega_1 / \|\omega_1\|, & u_2 &= \omega_2 - e_1 (e_1^T \omega_2), \\ e_2 &= u_2 / \|u_2\|, \\ e_3 &= e_1 \times e_2, \\ r_n &= (e_1, e_2, e_3), \\ m_n &= (C_\alpha)_n, \\ T_n &= (r_n, m_n). \end{aligned} \tag{17}$$

In this construction, two directional vectors are first defined: from $C_\alpha$ to C and $C_\alpha$ to N. We normalize the first direction $\boldsymbol{\omega}_1$ to define the local x-axis $\boldsymbol{e}_1$, and orthogonalize and normalize $\boldsymbol{\omega}_2$ to define the y-axis $\boldsymbol{e}_2$. The z-axis $\boldsymbol{e}_3$ is computed as the cross product of $\boldsymbol{e}_1$ and $\boldsymbol{e}_2$, forming a right-handed orthonormal basis. The resulting frame $T_n$ placing the local frame at the $C_\alpha$ of residue $n$. To define the local frame for placing the oxygen atom, we begin with the residue's central frame $T_n$, then apply a secondary transformation: $T^\star_{\mathrm{psi}}(\psi_n) = (\boldsymbol{r}_x(\psi_n), \boldsymbol{m}_{\mathrm{psi}})$, where $\psi_n$ represents a rotation angle along x-axis. The transformation funtions are defined as:

$$\boldsymbol{r}_x(\psi) = \begin{pmatrix} 1 & 0 & 0 \\ 0 & \cos\psi & -\sin\psi \\ 0 & \sin\psi & \cos\psi \end{pmatrix}, \quad \boldsymbol{m}_{\mathrm{psi}} = (1.526, 0.0, 0.0). \tag{18}$$

This transformation represents a rotation around the x-axis (aligned with the bond from $C_\alpha$ to C) by an angle $\psi_n$, followed by a translation to the position of the carbon atom $C^\star$ in the idealized frame centered at $C^\star_\alpha$. The combined transformation $T_n \cdot T^\star_{\mathrm{psi}}(\psi_n)$ thus defines the final frame in which the idealized oxygen $O^\star$ is placed to obtain its global coordinate.

## B  Diffusion on the Toric Manifold

A torsion vector $\chi \in [0, 2\pi)^d$ naturally resides on a flat d-dimensional torus, which can be represented as the quotient space $\mathbb{R}^d/L$, where $L = (2\pi\mathbb{Z}^d)$ denotes a discrete lattice subgroup of $\mathbb{R}^d$ isomorphic to $\mathbb{Z}^d$. This space models periodic angular data, and inherits a flat metric from its covering Euclidean space. The tangent space of the torus at any point is identified with $\mathbb{R}^d$, and all operations are performed modulo $2\pi$.

## C  Diffusion on SE(3)

Following previous work [58], we treat the group $SE(3)$ as the product space $SO(3) \times \mathbb{R}^3$, and endow it with a product Riemannian metric. Specifically, for tangent vectors $(a, b), (a', b') \in \mathrm{T}_r SO(3) \times \mathbb{R}^3$, the metric is defined as:

$$\langle (a, b), (a', b') \rangle_{SE(3)} = \langle a, a' \rangle_{SO(3)} + \langle b, b' \rangle_{\mathbb{R}^3}.$$

This structure allows for a natural decomposition of differential geometric objects on $SE(3)$ into rotational and translational components. In particular, the gradient of a function $f : SE(3) \to \mathbb{R}$ at $\mathcal{T} = (\boldsymbol{r}, \boldsymbol{x})$ is given by:

$$\nabla_{\mathcal{T}} f(\mathcal{T}) = [\nabla_r f(\boldsymbol{r}, \boldsymbol{m}), \nabla_m f(\boldsymbol{r}, \boldsymbol{m})],$$

and the Laplace–Beltrami operator decomposes as:

$$\Delta_{SE(3)} f(T) = \Delta_{SO(3)} f(\boldsymbol{r}, \boldsymbol{m}) + \Delta_{\mathbb{R}^3} f(\boldsymbol{r}, \boldsymbol{m}).$$

We define Brownian motion on $SE(3)$ as the product of independent Brownian motions on $SO(3)$ and $\mathbb{R}^3$:

$$\boldsymbol{B}_t^{SE(3)} = [\boldsymbol{B}_t^{SO(3)}, \boldsymbol{B}_t^{\mathbb{R}^3}]$$

where the rotational and translational components evolve independently. This product metric allows us to treat the rotational and translational components of the forward diffusion process independently, leading to the following decomposition of the conditional score:

$$\nabla_{\mathcal{T}_t} \log p_{t|0}(\mathcal{T}_t|\mathcal{T}_0) = [\nabla_{r_t} \log p_{t|0}(\boldsymbol{r}_t|\boldsymbol{r}_0), \nabla_{m_t} \log p_{t|0}(\boldsymbol{m}_t|\boldsymbol{m}_0)]$$

The forward process on $SE(3)$ is thus described by two independent SDEs. Let $\mathcal{M}$ be a compact Lie group (e.g., $SO(3)$), and let $\chi_\ell$ denote the character of the $\ell$-th irreducible unitary representation of dimension $d_\ell$. Then, the heat kernel (transition density of Brownian motion) on $\mathcal{M}$ is given by:

$$p_{t|0}(x_t|x_0) = \sum_{\ell \in \mathbb{N}} d_\ell e^{-\lambda_\ell t/2} \chi_\ell((x_0)^{-1} x_t).$$

where $\lambda_\ell$ is the eigenvalue of the Laplace–Beltrami operator associated with $\chi_\ell$. Specializing to $SO(3)$, the heat kernel becomes the isotropic Gaussian on $SO(3)$:

$$f(\omega, t) = \sum_{\ell \in \mathbb{N}} (2\ell + 1) e^{-\ell(\ell+1)t/2} \frac{\sin((\ell+1/2)\omega)}{\sin(\omega/2)}. \tag{19}$$

where $\omega$ is the angle of the relative rotation. The corresponding score function is:

$$\nabla \log p_{t|0}(\boldsymbol{r}_t \mid \boldsymbol{r}_0) = \tfrac{\boldsymbol{r}_t}{\omega_t} \log(\boldsymbol{r}_0^\top \boldsymbol{r}_t) \frac{\partial_\omega f(\omega^{(t)}, t)}{f(\omega^{(t)}, t)}, \tag{20}$$

where $\omega_t$ is the angle of the relative rotation $\boldsymbol{r}_0^\top \boldsymbol{r}_t$, and the matrix logarithm term maps to the tangent space at $\boldsymbol{r}_t$.

For the translational component, we model the forward process using a Variance Preserving SDE (VP-SDE). The transition density is given by:

$$p_{t|0}(\boldsymbol{m}_t|\boldsymbol{m}_0) = \mathcal{N}(x_t; e^{-t/2}\boldsymbol{m}_0, (1 - e^{-t})\,\mathrm{I}_3). \tag{21}$$

Then we can get the score as:

$$\nabla \log p_{t|0}(\boldsymbol{m}_t|\boldsymbol{m}_0) = \frac{1}{1 - e^{-t}}(e^{-t/2}\boldsymbol{m}_0 - \boldsymbol{m}_t). \tag{22}$$

We use a learned denoising network to approximate the conditional score of the full SE(3) transformation. The score is decomposed into rotational and translational components as follows:

$$\begin{aligned}
\nabla_{\mathcal{T}_t} \log p_{t|0}(\mathcal{T}_t \mid \hat{\mathcal{T}}_0) &= (s_\theta^{\boldsymbol{r}}(t, \mathcal{T}_t), s_\theta^{\boldsymbol{m}}(t, \mathcal{T}_t)), \\
s_\theta^{\boldsymbol{r}}(t, \mathcal{T}_t) &= \nabla_{\boldsymbol{r}_t} \log p_{t|0}(\boldsymbol{r}_t|\hat{\boldsymbol{r}}_0), \\
s_\theta^{\boldsymbol{m}}(t, \mathcal{T}_t) &= \nabla_{\boldsymbol{m}_t} \log p_{t|0}(\boldsymbol{m}_t|\hat{\boldsymbol{m}}_0).
\end{aligned} \tag{23}$$

## D   Architecture

Here we provide mathematical detail of PepBridge presented in method section. Let $\mathbf{h}^\ell = [h_1^\ell, \ldots, h_N^\ell] \in \mathbb{R}^{N \times D_h}$ denote the node embeddings at layer $\ell$, where $h_n^\ell$ represents the embedding for residue $n$. Similarly, let $\mathbf{z}^\ell \in \mathbb{R}^{N \times N \times D_z}$ represent the edge embeddings, where $z_{ij}^\ell$ encodes the interaction between residues $i$ and $j$.

Node embeddings are initialized using residue indices, atom coordinates, backbone dihedral angles, side-chain angles $h_\mathcal{B}$, and the diffusion timestep $t$. For edge (residue-pair) embeddings, we incorporate a combination of residue-type pairs, relative sequence positions, pairwise distances, and relative orientations. Each of these features is individually encoded using a dedicated multi-layer perceptron (MLP). The resulting feature vectors are concatenated and passed through another MLP to produce the final embeddings.

The initial layer-0 embeddings for residues $i$ and residue pairs $(i, j)$ are computed using MLPs applied to sinusoidal encodings $\phi(\cdot)$ of the input features:

$$\begin{aligned}
h_i^0 &= \mathrm{MLP}([\phi(h_{\mathcal{B}_i}), \phi(t)]) \in \mathbb{R}^{D_h}, \\
z_{ij}^0 &= \mathrm{MLP}([\phi(h_{\mathcal{B}_i}), \phi(h_{\mathcal{B}_i}), \phi(i - j), \phi(dis(i, j)), \phi(ori(i, j)), \phi(t)]) \in \mathbb{R}^{D_z},
\end{aligned} \tag{24}$$

where $D_h, D_z$ denote the dimensions of the node and edge embeddings, respectively. The functions $\phi(\mathrm{dis}(i, j))$ and $\phi(\mathrm{ori}(i, j))$ represent sinusoidal encodings of the distance and relative orientation between residues $i$ and $j$.

To encode the surface of the receptor protein, we extract node-level features from the surface points and apply an MLP to obtain embeddings. Each surface node is represented by its 3D position (surf$_t$), hydrogen bonding potential (surf$_{\mathrm{hbond}}$), and hydrophobicity score (surf$_{\mathrm{hp}}$). These features are concatenated and passed through an MLP to produce the surface node embeddings:

$$h_{\mathrm{surf}} = \mathrm{MLP}([\mathrm{surf}_t, \mathrm{surf}_{\mathrm{hbond}}, \mathrm{surf}_{\mathrm{hp}}]). \tag{25}$$

For the peptide surface representation, we encode only the 3D positional coordinates using an MLP, omitting auxiliary features such as hydrogen bonding and hydrophobicity. At each diffusion timestep $t$, the model takes as input the receptor's node and edge embeddings, the noised peptide descriptors, and a sinusoidal embedding of the timestep. It predicts a denoising score that guides the reverse diffusion process toward the clean peptide descriptors at $t = 0$. The model architecture is based on Invariant Point Attention (IPA), which employs SE(3)-invariant attention to capture interactions between the receptor and the peptide. The output of the IPA module is passed through separate MLP decoders to reconstruct various ground-truth peptide descriptors, such as atom coordinates, dihedral angles, and residue types. Notably, certain residue types may be partially inferred from the number of side-chain dihedral angles, due to structural constraints.

# E Experimental Details

The experiments were conducted on a computing cluster with 2 NVIDIA RTX A6000, each with 48 GB of memory. The total computation time for training was approximately 21 hours. We trained for 900000 steps with batch size 8. We used the Adam optimizer with a start learning of 5e-4. We also schedule to decay the learning rate exponentially with a factor of 0.6 and a minimum learning rate of 1e-6. The learning rate is decayed if there is no improvement for the validation loss in 10 consecutive evaluations.

## E.1 Dataset

The filtered dataset underwent sequence-based clustering using MMseqs2 [45], resulting in 9,816 protein-peptide complexes organized into 292 distinct clusters. For systematic evaluation, we designated 10 clusters encompassing 158 complexes as the test set, with the remaining complexes allocated to training and validation cohorts.

## E.2 Baselines

We briefly summarize the baselines and tools used in our study, including generative approaches for protein and peptide design, as well as side-chain packing methods.

- **ProteinGenerator** [27] is a RoseTTAFold-based diffusion model that jointly generates protein sequences and structures, with flexible conditioning on target sequence and structural attributes.
- **RFDiffusion** [50] is a generative model fine-tuned on structure denoising tasks, enabling high-accuracy design of monomers, binders, and symmetric protein assemblies.
- **PPFLOW** [26] is a target-aware peptide designer that performs conditional flow matching on torus manifolds to model peptide torsion-angle geometry.
- **PepFlow** [25] is a multimodal flow-matching model for full-atom peptide design targeting specific protein receptors. It jointly models backbone geometry, side-chain conformations, and residue identities over appropriate geometric manifolds.
- **PepGLAD** [21] combines geometric latent diffusion with receptor-specific transformations to generate full-atom peptides. The model operates in a learned latent space and adapts to diverse binding geometries for improved generalization.
- **Chroma** [19] is a unified generative framework for proteins and protein complexes that integrates a polymer-aware diffusion process with a scalable architecture, supporting constraint-driven design across sequence, structure, and function.
- **SCWRL4** [23] is a widely used side-chain packing tool employing a backbone-dependent rotamer library and a statistical energy function.
- **DLPacker** [34] is a 3D CNN-based model for residue side-chain conformation prediction. We utilize the official implementation along with the model weights.
- **AttnPacker** [32] utilizes equivariant attention mechanisms on backbone 3D geometry to predict all side-chain coordinates simultaneously.
- **DiffPack** [61] is a diffusion-based generative model that autoregressively samples side-chain angles on a toric manifold.

## E.3 Training Metrics Details

**RMSD.** Root-Mean-Square Deviation is a widely used metric for assessing structural similarity between proteins. In our evaluation, we align the generated peptide to the native peptide within the complex using the Kabsch algorithm. considering only the peptide portion for superposition. We then compute the RMSD based on normalized $C_\alpha$ atom distances between the generated and native peptides. Lower RMSD values indicate greater structural similarity.

**BSR.** Binding Site Recovery measures the similarity of interaction patterns between the generated and native peptide-protein complexes. Specifically, it evaluates whether the generated peptide

engages target protein residues in a manner similar to the native peptide, potentially reflecting similar biological functions. A residue in the protein is considered part of the binding site if its $C_\beta$ atom lies within 6 Å of any residue in the peptide. BSR is defined as the ratio of overlapping binding site residues between the generated and native complexes. Higher BSR values indicate greater similarity in binding interactions.

**Consistency** represents the statistical association between the clustering results of surface and structures. This metric quantifies how well a model captures the fundamental consistency between surfaces and their corresponding structures. A model that accurately represents the joint distribution should achieve a high score, while a low score indicates the model generates inconsistent surface-structure pairs. The evaluation process involves clustering both surfaces and structures, assigning discrete labels to each. These clustering labels can be interpreted as nominal variables. Given that similar surfaces should correspond to similar structures. We employ Cramér's V association [5] to measure this correlation, where a value of 1.0 indicates perfect association and 0.0 indicates no association. For surface representation, we first obtain molecular fingerprints using the methodology described in [44]. These fingerprints serve as input for the clustering algorithm, which assigns labels to the generated peptide surfaces.

**Diversity.** To assess diversity, we compute all pairwise TM-scores among the generated peptides for a given target using the original TM-align program. TM-scores quantify structural similarity between peptide pairs. We define diversity as 1 minus the average TM-score, where higher values indicate greater structural variability among the generated peptides. This metric reflects the breadth of structural exploration achieved during the design process.

### E.4 Hyperparameters

Our proposed method incorporates several hyperparameters, including sample step, learning rate, batch size and feature dimensions. To validate these hyperparameters, we conducted a random search. The search space are presented in Table 4.

Table 4: Search space for all PepBridge. The parameters used in validation are marked in **bold**

| Parameter | Search Space |
|---|---|
| Learning rate | 0.0009, 0.0007, **0.0005**, 0.00001 |
| Hidden dimension of residue feature | 64, **128**, 256 |
| Hidden dimension of edge feature | **64**, 128, 256 |
| Hidden dimension of surface feature | **16**, 24, 32 |
| Number of attention heads | **8**, 16, 24 |
| Loss weight of surface | 0.1, **0.5**, 1 |
| Loss weight of backbone position | 0.1, 0.5, **1** |
| Loss weight of backbone rotation | 0.1, 0.5, **1** |
| Sampling steps | 500, **1000**, 1500 |
| Training steps | 500, **1000**, 1500 |
| Batch size | 4, **8**, 16 |

### E.5 Computational Complexity

We compared PepBridge with several baseline methods in terms of training time, inference time per sample, GPU usage, and model size. The time cost is reported as the total time spent divided by the number of designed candidates. As summarized in Table 5, multi-modal processing does introduce additional complexity relative to uni-modal approaches, our analysis shows that PepBridge achieves a good balance between computational efficiency and performance.

Table 5: Computational cost and resource footprint across methods. Training and inference times are normalized per designed sample.

| Method | Training time | Inference time (s/sample) | GPU(s) used | Params (M) |
|---|---|---|---|---|
| RFdiffusion | 3 days | 80–180 | 8×A100 | ∼120 |
| ProteinGenerator | 4 weeks | 152 | 64×V100 | ∼120 |
| PepFlow | 20 hours | 14–24 | 2×A6000 | ∼7 |
| Chroma | 10 weeks | 185–226 | 8×V100 | ∼20 |
| PepGLAD | 29 hours | 3 | 1×24 GB GPU | ∼2.5 |
| PepBridge | 21 hours | 16–37 | 2×A6000 | ∼10 |

# F   Additional Experiments

## F.1   Visualization

Figure 1 provides additional examples of peptides generated by PepBridge. These visualizations include both surface and backbone structures of the generated peptides in a top-down view, further illustrating the model's ability to produce geometrically coherent and interface-aware designs.

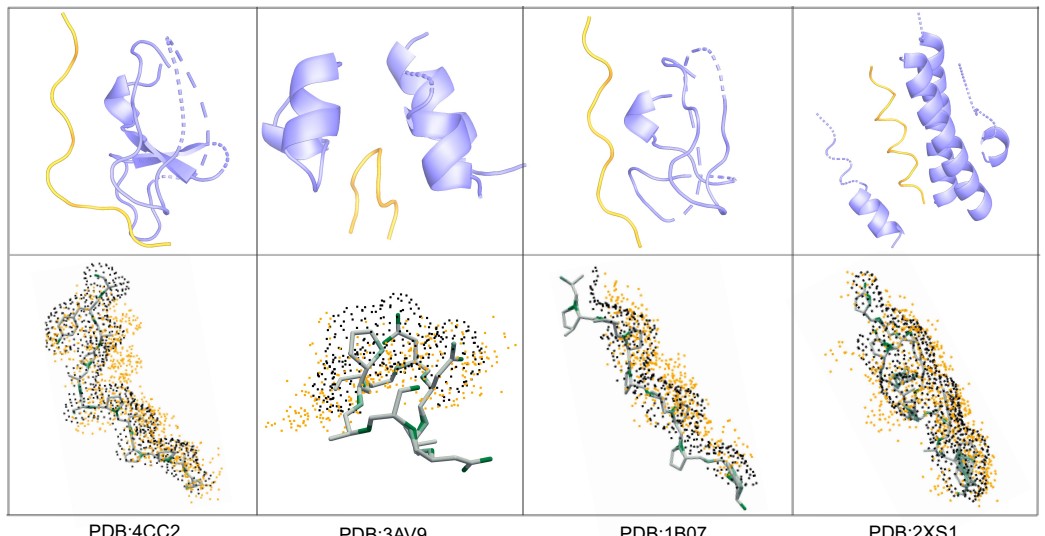

| PDB:4CC2 | PDB:3AV9 | PDB:1B07 | PDB:2XS1 |

Figure 4: Visualization of generated peptides by PepBridge. **Top:** Generated peptides (in orange) for receptors (in purple). The PDB ID of the receptors are 4CC2, 3AV9, 1B07, and 2XS1. **Bottom:** The generated backbone structure and surface. The ground-truth surface structure (in black) and generated surface (in orange) are shown to compare the ability of interface caption.

## F.2   Training and Sampling Time Steps

We ablated the number of diffusion steps used during training while fixing the sampling procedure at 1000 steps. The results are summarized in Table 6. Increasing the number of steps consistently improved all quality metrics. The largest gains occurred up to 1000 steps, with smaller but still measurable improvements between 1000 and 1500 steps. Given these trends, we adopt 1000 training steps as a favorable trade-off between overall accuracy and computational cost.

We further conducted experiments to assess how different inference time steps affect the final performance metrics using the model trained with 1000 steps. As shown in Table 7, the results indicate that longer sampling chains (e.g., 1000 steps) improve the quality of the generated structures. However, performance gains begin to plateau beyond 800 steps, and we observe a slight decrease in structural diversity, suggesting a trade-off between generation determinism and diversity. Based on this analysis, we selected 1000 steps as the default setting to achieve the best overall balance.

Table 6: Effect of training steps on performance (sampling fixed at 1000). Higher is better (↑) except RMSD (↓).

| | $\text{Div}_{\text{stru}}$ (↑) | Aff. % (↑) | Stab. % (↑) | RMSD Å(↓) | BSR (↑) |
|---|---|---|---|---|---|
| PepBridge (time step =500) | 0.57 | 18.96 | 24.79 | 2.96 | 82.84 |
| PepBridge (time step =800) | 0.61 | 19.07 | 26.31 | 2.36 | 85.57 |
| PepBridge (time step =1000) | 0.59 | 19.16 | 25.02 | 2.19 | 83.90 |
| PepBridge (time step =1500) | 0.62 | 23.28 | 26.68 | 2.11 | 86.92 |

Table 7: Effect of sampling steps on performance (model trained with 1000 steps). Higher is better (↑) except RMSD (↓).

| | $\text{Div}_{\text{stru}}$ (↑) | Aff. % (↑) | Stab. % (↑) | RMSD Å(↓) | BSR (↑) |
|---|---|---|---|---|---|
| PepBridge (time step =500) | 0.61 | 17.42 | 23.97 | 2.85 | 80.05 |
| PepBridge (time step =800) | 0.62 | 18.77 | 24.58 | 2.69 | 82.33 |
| PepBridge (time step =1000) | 0.59 | 19.16 | 25.02 | 2.19 | 83.90 |
| PepBridge (time step =1500) | 0.56 | 18.46 | 24.71 | 2.74 | 83.78 |

# G   Limitations and Future Work

While PepBridge presents a structured approach to joint protein surface and backbone design, several limitations remain that can be addressed in future work. One key limitation lies in the simplification of surface representations. The current model relies on solvent-accessible point clouds with biochemical annotations, which, while effective, may not fully capture finer molecular interactions such as electrostatic potential fields and solvent dynamics. These factors play crucial roles in protein-protein interactions and could enhance the accuracy of designed peptides if incorporated. Another limitation is the model's reliance on receptor geometry. PepBridge assumes that receptor surface features sufficiently dictate the constraints on peptide binding. However, this does not account for receptor flexibility or conformational changes upon ligand binding, which are common in many biological systems. Addressing this aspect could make the model more applicable to highly dynamic binding sites. Computational efficiency also poses a challenge. The diffusion bridge model and SE(3) diffusion backbone generation require computationally intensive sampling. While the hierarchical generation process improves efficiency, generating high-quality peptides remains expensive, particularly for longer sequences. Further optimization is necessary to reduce the computational cost while maintaining or improving accuracy. Additionally, the current evaluation primarily focuses on geometric complementarity and binding affinity predictions. While these provide useful insights, they do not fully capture the functional stability of designed peptides. Wet-lab experiments and molecular dynamics simulations are necessary to assess real-world applicability, ensuring that the generated structures remain stable under physiological conditions.

Future work can address these limitations in several ways. Enhancing surface representations by incorporating higher-order biochemical features such as electrostatic potential fields, solvent effects, or graph-based molecular embeddings could improve the precision of surface-conditioned peptide generation. Furthermore, integrating receptor flexibility into the model by leveraging conformational ensembles or reinforcement learning-based refinement strategies would allow for more realistic modeling of dynamic binding sites. To improve computational efficiency, future research could explore accelerated sampling techniques, such as adaptive noise schedules, score distillation sampling (SDS), or flow-matching approaches. These methods could significantly reduce inference time while preserving or enhancing model accuracy. Finally, validating PepBridge through real-world applications, particularly in therapeutic peptide design, remains a crucial next step. Incorporating experimental validation through biochemical assays and integrating co-evolutionary signals into the design process could further enhance the biological relevance of the generated structures. By addressing these challenges, PepBridge can be refined to enable more accurate, efficient, and versatile protein design.

