# OpenReview forum: "Joint Design of Protein Surface and Backbone Using a Diffusion Bridge Model"
_NeurIPS.cc/2025/Conference — NeurIPS 2025 poster_

### Official Review · Reviewer_Q8ka · 2025-06-27

**Clarity:** 3
**Significance:** 3
**Originality:** 3
**Rating:** 3
**Confidence:** 4

**Summary:**

This paper proposes a top-down joint design framework for protein surface and structure based on a multi-modal diffusion approach. The key innovation lies in employing a denoising diffusion bridge model to directly map the receptor surface to the ligand surface, rather than relying on a traditional Gaussian noise prior.

**Questions:**

1. How do different hyperparameters affect the final performance?

2. Although diversity metrics have been reported, there is a lack of in-depth analysis on how to find the optimal balance between diversity and quality.

3. Although visual results are provided, I cannot see if they meet the practical significance. Can it be used for practical applications? Does the author plan to conduct wet experiments for validation? How to prove the reliability of computational predictions without experimental verification?

4. The multi-modal diffusion process requires simultaneous processing of surfaces, torsion angles, and residue types, which inevitably leads to computational overhead. Is the additional computational cost reasonable compared to existing uni-modal methods? Can you provide a detailed analysis of computation time and resource consumption?

5. Is it applicable to receptors without obvious binding pockets?

6. There is another small but important point: the introduction of surface modeling is an innovative aspect of the paper, but how can it be proven to be effective?

**Ethical Concerns:**

["NO or VERY MINOR ethics concerns only"]

**Final Justification:**

I have read the feedback from other reviewers. I hope the author can respond to my additional questions before I consider maintaining my score. In addition, after further discussion, I found that the author's experimental results showed that performance actually declined as the timestamp increased. This is a counterintuitive result. As the timestamp increases, the model's performance tends to saturate and stabilise. Performance will continue to improve, even if the improvement is limited, but I do not believe that it will decline. I am concerned about the author's experimental setup. Based on this, I decided to lower my score.

Latest update (AOE 8.8): The authors further clarify that the model was trained at 1000 timestamps and tested at other numbers of timestamps. I think this contradicts the author's previous response. (The author claimed in the previous rebuttal that "Based on this analysis, we selected 1000 steps as the default setting to achieve the best overall balance". Therefore, I am not sure that the reason why the author is able to make such a judgment is that he has trained on all timestamps.) If the results provided by the author are accurate, then the experimental setup is incorrect. So I am very uncertain whether the experiment provided by the author is accurate. If my judgment is incorrect, I am willing to increase my score to 4. Please AC confirm further.

**Limitations:**

yes

**Paper Formatting Concerns:**

The format of the paper is well organized.

**Quality:**

2

**Strengths And Weaknesses:**

Strengths:
1. Using the receptor surface as an informative prior instead of Gaussian noise is a promising approach.

2. The methodological design is well-founded, decomposing the complex protein design problem into two interrelated sub-tasks: surface generation and structure prediction.

3. The topic is very innovative, although the method has certain shortcomings compared to existing methods.

Weaknesses:
1. In the introduction, the author mentions "(i) Limited ability to generate diverse ..." However, according to the experimental results in Table 1, it seems that the "Div" metric has not improved compared to "PepFlow w/Bb", but rather decreased.

2. In the original paper of "PepFlow w/Bb", the effect of "PepFlow w/Bb+Seq+Ang" seems to be the best, why not compare it with "PepFlow w/Bb+Seq+Ang"? Although there have been improvements in some metrics, they do not show a breakthrough compared to "PepFlow w/Bb".

3. The paper did not provide a detailed analysis of computational complexity and comparison of running time. The multi-modal diffusion process may bring significant computational overhead.

4. The figures in the paper are beautifully drawn, but it would be clearer to see if there were more textual descriptions.

5. It seems that the author may need to further compare with [1].

6. The PPFLOW [2] method has been mentioned in the related work section, but why hasn't it been compared in Table 1? Can you explain it?

[1] Hotspot-Driven Peptide Design via Multi-Fragment Autoregressive Extension. ICLR 2025

[2] PPFLOW: Target-Aware Peptide Design with Torsional Flow Matching. ICML 2024

---

> ### Author Rebuttal · Authors · 2025-07-30
>
> We appreciate the constructive feedback which helps us enhance the readability and quality of our paper. Below are our point-wise responses to the reviewer's concerns:
>
> ----------------------------------------
> > W1. The claim in the introduction that method addresses "Limited ability to generate diverse ..." while Table 1 shows their "Div" (diversity) metric decreased compared to baseline "PepFlow w/Bb" method.
>
> Thank you for your insightful comment. Our claim: "Limited ability to generate diverse yet receptor-compatible surface configurations" . This refers to the diversity of 3D molecular surface geometries at the protein-protein interface. Table 1 metric: The "Div" metric measures structural diversity using TM-score between generated peptide backbones, which captures overall fold similarity rather than surface-specific diversity. Our method generates peptide surfaces de novo, enabling exploration of novel surface configurations that may not be captured by backbone TM-score diversity. A peptide can have similar overall fold (high TM-score) but present very different surface properties for receptor binding. In addition, since structures with TM-scores >0.5 maintain recognizable folds while allowing local surface variations, this constrained diversity may actually be more biologically relevant than purely maximizing structural dissimilarity.
>
> > W2. Compare against “PepFlow w/Bb+Seq+Ang.
>
> Thank you for pointing this out. We include additional experiments comparing PepBridge variants directly against this method across all core metrics. Preliminary results show that “PepBridge w/Bb+Seq+Ang+Surf” achieves comparable or better performance in terms of stability, while "PepBridge w/Bb+Seq+surf" demonstrates better performance on affinity and diversity. We will include these results in Table 1 and expand the discussion accordingly.
>
> | | Div\_stru (↑) | Aff. % (↑) | Stab. % (↑) | RMSD Å (↓) | BSR (↑)|
> |:--|:-:|:----:|:---:|:---:|:----:|
> |PepFlow w/Bb                   |**0.64**   |18.10      |14.04      |2.30|82.17|
> |PepFlow w/Bb+Seq               |0.50       |19.39      |19.20      |2.21|_85.19_|
> |PepFlow w/Bb+Seq+Ang           |0.42       |_21.37_|18.15      |_2.07_|**86.89**|
> |PepBridge w/Bb+surf            |0.60       |20.07      |_21.75_|**2.04**|84.62|
> |PepBridge w/Bb+Seq+surf        |_0.62_ |**22.07**  |20.71      |2.18|84.91|
> |PepBridge w/Bb+Seq+Ang+surf    |0.59       |19.16      |**25.02**|2.19|83.90|
>
> > W4. Add more textual descriptions to figures for clarity.
>
> We appreciate this suggestion and have added more detailed textual explanations alongside key figures. The updated description for Figure 1 is: “In the top-down representation of a protein, the molecular surface plays a central role in mediating interactions between the peptide and its target receptor. Each residue is defined by its backbone and side-chain atoms, whose spatial positions are governed by a rotation $r$, a translation $m$, and torsional angles $\psi$ and ${\chi}$.”
>
> > w5 w6. Compare with recent methods PepHAR and PPFLOW in the main experiments.
>
> Thank you for raising this point. We conducted additional experiments to benchmark our PepBridge variants against both methods. We will update Table 1 accordingly and expand the discussion.
>
> | | Div\_stru (↑) | Aff. % (↑) | Stab. % (↑) | RMSD Å (↓) | BSR (↑)|
> |:--|:-:|:----:|:---:|:---:|:----:|
> |PepHAR (K=1)                   |0.58      |18.56      |15.69|3.73|84.17|
> |PepHAR (K=2)                   |0.61      |19.82      |15.91|3.19|84.57|
> |PepHAR (K=3)                   |**0.65**  |_20.53_|16.62|2.68|**86.74**|
> |PPFLOW                         |0.53      |17.62      | 17,25|2.94 | 78.72
> |PepBridge w/Bb+surf            |0.60      |20.07      |_21.75_|**2.04**|84.62|
> |PepBridge w/Bb+Seq+surf        |_0.62_|**22.07**  |20.71      |_2.18_|_84.91_|
> |PepBridge w/Bb+Seq+Ang+surf    |0.59      |19.16      |**25.02**|2.19|83.90|
>
> > Q1. How do different hyperparameters affect the final performance?
>
> Thank you for this important question. We conducted experiments to assess how different sampling time steps affect the final performance metrics. The results indicate that longer sampling chains (e.g., 1000 steps) improve the quality of the generated structures. However, performance gains begin to plateau beyond 800 steps, and we observe a slight decrease in structural diversity, suggesting a trade-off between generation determinism and diversity. Based on this analysis, we selected 1000 steps as the default setting to achieve the best overall balance.
>
> | | Div\_stru (↑) | Aff. % (↑) | Stab. % (↑) | RMSD Å (↓) | BSR (↑)|
> |:--|:-:|:----:|:---:|:---:|:----:|
> |PepBridge (time step =500) |0.61  |17.42 |23.97|2.85|80.05|
> |PepBridge  (time step =800)  |0.62 |18.77 |24.58|2.69|82.33|
> |PepBridge  (time step =1000) |0.59 |19.16 |25.02|2.19|83.90|
>
> > Q2. Although diversity metrics have been reported, there is a lack of in-depth analysis on how to find the optimal balance between diversity and quality.
>
> PepBridge addresses the diversity-quality trade-off through flexible sampling strategies. We explored different sampling approaches to tune this balance: DDPM sampling: Incorporates Gaussian noise at each denoising step, yielding higher diversity but potentially lower structural quality. DDIM sampling: Uses deterministic denoising trajectories, producing higher-quality structures with reduced diversity. This flexibility allows us to prioritize either exploration (diversity) or optimization (quality) based on specific design objectives. Future work could develop adaptive sampling strategies that dynamically adjust noise levels based on intermediate quality metrics, potentially achieving both high diversity and quality simultaneously.
>
> > Q3. The method’s practical applicability, plans for experimental validation, and the reliability of its computational predictions.
>
> Thank you for your insightful comment. While computational results show promising agreement with known structural and energetic metrics, experimental validation remains essential. We acknowledge these results represent necessary but not sufficient conditions for biological activity.
>
> We are planning to develop a multi-tiered validation approach:
>
> `1. In silico validation:`
>
> Molecular dynamics (MD) simulations and free energy perturbation (FEP) calculations on top-ranked candidates to assess binding stability and thermodynamics under physiologically relevant conditions.
>
> `2. Biochemical assays:`
>
> Wet-lab binding assays (e.g., surface plasmon resonance, isothermal titration calorimetry) to measure actual binding affinities and kinetics.
>
> `3. Enhancing Computational Reliability:`
>
> To improve prediction reliability, we plan to develop PepBridge-derived scoring functions for virtual screening applications.
>
> > W3 Q4. Computational complexity and comparison of running time.
>
> We appreciate the reviewer’s concern regarding the potential computational overhead introduced by our multi-modal diffusion framework. To address this, we compared PepBridge with several baseline methods in terms of training time, inference time per sample, GPU usage, and model size. The time cost is reported as the total time spent divided by the number of designed candidates. While multi-modal processing does introduce additional complexity relative to uni-modal approaches, our analysis shows that PepBridge achieves a good balance between computational efficiency and performance.
>
> ### Table: Comparison of Computation Time and Resources
>
> | Method   | Training Time  | Inference Time (s/sample) | GPU(s) Used |  Params (M) |
> |-------------------------------|---------------------|----------------------------|-------------------|------------|
> | RFdiffusion | 3 days | 80–180  | 8 A100 | ~ 120 |
> | ProteinGenerator | 4 weeks | 152 | 64 V100 | ~ 120 |
> | PepFlow | 20 hours | 14-24 | 2 A6000 | ~ 7 |
> | Chroma  | 10 weeks | 185-226 | 8 V100  | ~ 20 |
> | PepGLAD  | 29 hours | 3 | 24G GPU| ~ 2.5 |
> | PepBridge | 21 hours | 16-37 | 2 A6000  | ~ 10 |
>
> > Q5. The generation of peptides targeting receptors without obvious binding pockets.
>
> Thank you for this important question about PepBridge's applicability to receptors lacking well-defined binding pockets. We conducted experiments using randomly sampled surface patches to quantify this effect.
>
> | | Div\_stru (↑) | Aff. % (↑) | Stab. % (↑) | BSR (↑)|
> |:--|:-:|:----:|:---:|:---:|
> |PepBridge (random pockets) |0.61 |18.95 |25.17|84.12|
>
> These results demonstrate that PepBridge remains effective even when applied to less constrained or uncertain surface regions, maintaining high structural and biochemical performance.
>
> For receptors without obvious binding pockets, we recommend a two-step approach:
>
> Binding site prediction preprocessing: Users can integrate established binding site prediction tools to identify potential interaction regions before peptide generation
>
> Multi-site exploration: When binding sites are uncertain, PepBridge can be applied to multiple predicted or suspected interaction regions, allowing exploration of different binding hypotheses.
>
> > Q6. The effectiveness of incorporating surface modeling.
>
> We appreciate the reviewer’s question. We conducted ablation studies using PepBridge variants with and without surface representations. The results, shown in the table below, demonstrate that incorporating surface information leads to substantial improvements across multiple performance metrics.
>
> | | Div\_stru (↑) | Aff. % (↑) | Stab. % (↑) | RMSD Å (↓) | BSR (↑)|
> |:--|:-:|:----:|:---:|:---:|:----:|
> |PepBridge w/Bb+Seq         |0.56 |17.12  |16.41|2.87|79.79|
> |PepBridge w/Bb+Seq+surf    |0.62 |22.07  |20.71|2.18|84.91|
> |PepBridge w/Bb+Seq+Ang     |0.53 |18.46  |19.73|3.21|76.53|
> |PepBridge w/Bb+Seq+Ang+surf|0.59 |19.16  |25.02|2.19|83.90|
>
> ------------------------------
> We are grateful for your constructive feedback and thoughtful suggestions that helped us refine our work.

---

> ### Comment · Reviewer_Q8ka · 2025-08-02
> **Further questions**
>
> Thank you for responding to my points. Thank you for the additional experiments, but they have introduced additional questions.
>
> W2: Overall, PepFlow seems to have more advantages, especially Div_stru and BSR, which seem to have a significant gap compared to others. According to Q6, the introduction of the surface significantly improved performance. I am curious about how PepFlow performs after introducing surface modeling.
>
> Q1: In my opinion, as the time step gradually increases, the performance of the model should gradually stabilize. However, the results show that the performance of the model is still continuously improving. Moreover, according to the comparison results in Table 1, the magnitude of improvement is quite significant. I am curious about the upper limit.
>
> Q3: Could the author provide more analysis and discussion on the generated results compared to the protein surface used in the real world?

---

> > ### Author Response · Authors · 2025-08-05
> > **Reply to Updated Comment 1**
> >
> > > N W2. Performance Comparison and Integration of Surface Modeling in PepFlow.
> >
> > Thank you for your insightful question.
> > Div_stru is calculated based on one minus the pairwise TM-score between generated structures.
> > TM-scores below 0.5 typically indicate non-native-like or unreliable folds. Structures in the 0.3–0.5 range may reflect potentially novel folds—but they require cautious interpretation and validation. Conversely, TM-scores above 0.5 represent conservative innovations that preserve global fold characteristics while allowing meaningful local variation.
> > Thus, the model should balance between generating diverse structures and maintaining structural plausibility. In practice, affinity remains the most critical metric for assessing the utility of generated proteins, especially in functional contexts.
> >
> > In response to the reviewer’s question about whether PepFlow could benefit from incorporating surface context similarly to methods discussed. we believe the answer is yes.
> > Both flow matching and diffusion bridge models aim to model transitions between two distributions, but they do so via different mechanisms. Diffusion models use stochastic forward and reverse processes, while flow matching directly models continuous deterministic flows using learned vector fields.
> > Incorporating surface context enriches the conditioning space in both paradigms, allowing the model to generate protein that are not only structurally plausible but also better optimized for surface complementarity. We view surface-aware generation as a highly promising direction and are actively exploring how to integrate such context into PepFlow’s framework in future work.
> >
> > > N Q1.  Impact on performance with increasing time steps.
> >
> > Thank you for your insightful observation. Our model is trained with a fixed number of 1000 time steps, and we have found that performance does not improve beyond this point. In fact, increasing the number of steps beyond 1000 can degrade the quality of the generated samples. As shown in the table below, such increases lead to diminishing returns. This degradation is likely caused by a misalignment between the noise schedule used during sampling and the one used during training, which can impair the model’s ability to generate high-quality outputs and introduce artifacts such as over-smoothing and loss of structural detail.
> >
> > | | Div\_stru (↑) | Aff. % (↑) | Stab. % (↑) | RMSD Å (↓) | BSR (↑)|
> > |:--|:-:|:----:|:---:|:---:|:----:|
> > |PepBridge  (time step =1000) |0.59 |19.16 |25.02|2.19|83.90|
> > |PepBridge  (time step =1200)  |0.58 |19.14 |25.13|2.59|84.71|
> > |PepBridge  (time step =1500) |0.56 |18.46 |24.71|2.74|83.78|
> >
> > > N Q3: Further analysis on the comparison of protein surfaces.
> >
> > We thank the reviewer for the insightful suggestion. To analyze and discuss the similarity between the generated molecular surfaces and ground-truth protein surfaces, we follow the evaluation protocol introduced in [1] and adopt three metrics from 3D shape modeling: Volumetric Intersection over Union (IoU), Chamfer Distance (CD), and Normal Consistency (NC). These three metrics are all normalized to a range of 0-1. These metrics capture complementary aspects of surface similarity, including spatial overlap, geometric distance, and orientation alignment.
> >
> > 1. Volumetric Intersection over Union (IoU):
> >
> > This metric quantifies the volumetric overlap between the predicted and ground-truth molecular surfaces. It is defined as:
> >
> > $$
> > \text{IoU}(A, B) = \frac{|A \cap B|}{|A \cup B|}
> > $$
> >
> > where $A$ and $B$ denote the voxelized volumes of the predicted and target protein surfaces, respectively. Higher IoU values indicate more accurate spatial correspondence and better global surface reconstruction.
> >
> > 2. Chamfer Distance (CD):
> >
> > CD measures the average closest-point distance between two point clouds representing the surfaces. For two point sets $X_1 ,X_2$,
> >
> > $$
> > d_C(X_1, X_2) = \frac{1}{2} \left( \frac{1}{|X_1|} \sum_{x_1 \in X_1} \min_{x_2 \in X_2} \|x_1 - x_2\| + \frac{1}{|X_2|} \sum_{x_2 \in X_2} \min_{x_1 \in X_1} \|x_2 - x_1\| \right)
> > $$
> >
> > Lower values reflect closer geometric alignment.
> >
> > 3. Normal Consistency (NC):
> >
> > NC evaluates the alignment between surface normals of predicted and ground-truth surfaces. It is computed as:
> >
> > $$
> > \Gamma(X_{\text{gt}}, X_{\text{pred}}) = \frac{1}{|X_{\text{gt}}|} \sum_{j \in |X_{\text{gt}}|} \vec{n}_j \cdot \vec{m}
> > $$
> >
> > $$
> > \theta(y_j, X_{\text{pred}}) = \arg\min_{i \in |X_{\text{pred}}|} \| y_j - x_i \|_2
> > $$
> >
> > where $ \vec{m} $ is parametered by ${\theta(y_j, X_{\text{pred}})}$. A higher NC implies better preservation of local surface orientation.
> >
> > The results are summarized below. These results demonstrate that PepBridge can generate molecular surfaces that closely approximate real-world protein surfaces.
> >
> > | | IoU (↑) | Chamfer_dist (↓) | NC (↑) |
> > |:--|:-:|:----:|:---:|
> > |PepBridge |0.89 |0.14 |0.76|
> >
> > [1]. DSR: Dynamical Surface Representation as Implicit Neural Networks for Protein. NeurIPS 2023.

---

> > > ### Comment · Reviewer_Q8ka · 2025-08-06
> > > **Question about Q1**
> > >
> > > I thank the authors for their response. It is interesting that as the time stamp increases, the performance gradually decreases; can you explain that aspect a little bit better? Shouldn't an increase in time stamp mean finer sampling?

---

> ### Author Response · Authors · 2025-08-08
> **Reply to Updated Comment 2**
>
> Thank you for your insightful comment. From the results shown in our chart, when the number of inference time steps exceeds the number used during training, BSR and Stability show a slight increase followed by a small decrease, while Div_stru fluctuates within a reasonable range. The performance of the affinity metric declines. We intend to refine sampling granularity and scaling to better capture detailed performance patterns.
>
> Ma et al. [1] investigated inference-time scaling for diffusion models, reporting performance metrics such as FID and IS on ImageNet, and CLIPScore and Aesthetic Score on DrawBench (Figure 1 in [1]). The results show that some metrics (e.g., Aesthetic Score) may decline when inference time exceeds the training regime, despite initial improvements. In our case, we hypothesize that increasing the inference time steps beyond the training regime can lead to over-smoothing and loss of fine structural details in the generated protein structures. This reduces binding affinity, even if stability remains relatively steady. While longer inference can provide more refined sampling in principle, excessive steps may degrade structural features, preventing a uniform performance plateau across all metrics. Although our study focuses on proteins in 3D space rather than images, the analogy with the image domain suggests that increased inference time is not beneficial across all quality measures. We believe that investigating inference time and inference-time scaling is an interesting direction for future work.
>
> [1] Ma N, Tong S, Jia H, Hu H, Su YC, Zhang M, Yang X, Li Y, Jaakkola T, Jia X, Xie S. Inference-time scaling for diffusion models beyond scaling denoising steps.

---

> > ### Comment · Reviewer_Q8ka · 2025-08-08
> >
> > Hmm, but I still hope that the experiment can be conducted under the same settings. Instead of training only 1000 timestamps and testing under other numbers of timestamps. And the claim "Based on this analysis, we selected 1000 steps as the default setting to achieve the best overall balance" is inaccurate.
> >
> > I thank the author for their feedback.

---

> > > ### Author Response · Authors · 2025-08-09
> > > **Reply to Updated Comment 3**
> > >
> > > Thank you for this excellent point. The table provided in our rebuttal presents sampling-only comparisons based on a model trained with a 1000-step schedule.
> > > Therefore, the conclusion that “1000 is the best default” should be restated as “best default sampling length for our model trained with a 1000-step schedule.”
> > >
> > > We agree that it is important to determine whether 1000 remains the best default when the training and evaluation time steps are matched. Due to the limited time available during the discussion period, we were unable to conduct these additional experiments; however, we will include results varying both the training time steps and the sampling time steps in our refined manuscript.

---

### Official Review · Reviewer_WAQ2 · 2025-06-30

**Clarity:** 3
**Significance:** 3
**Originality:** 3
**Rating:** 5
**Confidence:** 3

**Summary:**

The authors introduce an approach that uses an annotated 3D surface of a receptor (with amino acid and atom position information) and generate a compatible ligand surface as well as the backbone in the form of frame geometry. They use diffusion bridge models starting from the receptor surface instead of Gaussian noise, which they demonstrate to be beneficial. As for the backbone generation, they use a surface frame matching network which is jointly trained. With their approach, they generate high quality samples and also demonstrate superior side-chain packing performance.

**Questions:**

1. I would be interested in seeing how consistent the results are, as the authors did not provide any standard deviation. In the caption of table 1 they state that they draw 40 candidates, but it is hard so say if this is enough for the results to be consistent. Also, did the authors run the inference with multiple different seeds?

2. How does the runtime of the different methods compare, and how does it scale with size? How many inference steps did have been used for the diffusion model and how does the number of steps impact the sample quality?

3. Although I am aware that diffusion bridges can be seen as a generalization of flow matching, I am curious to hear why the authors decided opted to use diffusion bridges instead of flow matching?

4. It is not entirely clear to me how the authors encoded the surface structure. As I understood it, they use a point cloud for the surface, but I am not sure how many points do they use (one for each atom?) and if it could benefit from a more precise parameterization (i.e., more points).

**Ethical Concerns:**

["NO or VERY MINOR ethics concerns only"]

**Final Justification:**

The authors present a novel framework in a well-prepared manuscript. PepBridge shows strong empirical results, with clear visualizations and a comprehensive set of experiments demonstrating its practical usefulness.

During the rebuttal phase, the authors addressed my main concerns, including the consistency of results by providing standard deviations and clarifying the number of inference samples. They also elaborated on architectural choices (e.g., diffusion bridges vs. flow matching vs. stochastic interpolants) and provided further implementation details. Looking at the other reviews, the updated manuscript will include new ablation studies that clarify the contribution of individual components.

Overall, I believe that the paper is well-motivated, the approach original, and the experiments support the claims.

**Limitations:**

yes

**Paper Formatting Concerns:**

No major concerns.

**Quality:**

3

**Strengths And Weaknesses:**

The paper is well written, clearly structured and easy to follow. The authors have presented (to my knowledge) novel ideas and new models; especially the idea of incorporating the surface structure seems promising. This idea could potentially transfer to other applications and be used in a variety of different models and settings.

The authors show improved results in some metrics (such as the RMSD), but do not improve the results in all benchmarks. Due to the lack of error bars and/or standard deviations, it is hard to judge how consistent the improvements are. Especially because most models are intrinsically stochastic during inference.

However, the authors mitigate this by comparing their approach with multiple different methods and also perform ablation studies how the different parts of their approach impact the main results. Compared to other approaches, their model support side-chain packing, making it more versatile.

---

> ### Author Rebuttal · Authors · 2025-07-30
>
> We appreciate the reviewer’s thoughtful feedback and suggestions, which have helped us improve the clarity and completeness of our work. Below, we address each of the concerns raised.
>
> > Q1. I would be interested in seeing how consistent the results are, as the authors did not provide any standard deviation. In the caption of table 1 they state that they draw 40 candidates, but it is hard so say if this is enough for the results to be consistent. Also, did the authors run the inference with multiple different seeds?
>
> Thank you for this insightful question. We indeed conducted inference using 5 different random seeds, generating 40 candidates per seed. The table below reports the standard deviation across all runs. These results demonstrate that the performance of our method is consistent across different random seeds.
>
> | | Div\_stru (↑) | Aff. % (↑) | Stab. % (↑) | RMSD Å (↓) | BSR (↑)|
> |:--|:-:|:----:|:---:|:---:|:----:|
> |PepBridge |0.59 $\pm$ 0.12|19.16 $\pm$ 0.87 |25.02 $\pm$ 1.36|2.19 $\pm$ 0.16|83.90 $\pm$ 1.61|
>
> > Q2. How does the runtime of the different methods compare, and how does it scale with size? How many inference steps did have been used for the diffusion model and how does the number of steps impact the sample quality?
>
> We appreciate the reviewer’s concern regarding the potential computational overhead introduced by our multi-modal diffusion framework. To address this, we compared PepBridge with several baseline methods in terms of training time, inference time per sample, GPU usage, and model size. The time cost is reported as the total time spent divided by the number of designed candidates.
>
> While multi-modal processing does introduce additional complexity relative to uni-modal approaches, our analysis shows that PepBridge achieves a strong balance between computational efficiency and performance. The modest increase in resource usage is well-justified by the substantial gains in structural precision and interface quality, especially in biologically critical tasks such as therapeutic protein design.
>
> ## Table: Comparison of Computation Time and Resources
>
> | Method   | Training Time  | Inference Time (s/sample) | GPU(s) Used |  Params (M) |
> |-------------------------------|---------------------|----------------------------|-------------------|------------|
> | RFdiffusion | 3 days | 80–180  | 8 A100 | ~ 120 |
> | ProteinGenerator | 4 weeks | 152 | 64 V100 | ~ 120 |
> | PepFlow | 20 hours | 14-24 | 2 A6000 | ~ 7 |
> | Chroma  | 10 weeks | 185-226 | 8 V100  | ~ 20 |
> | PepGLAD  | 29 hours | 3 | 24G GPU| ~ 2.5 |
> | PepBridge | 21 hours | 16-37 | 2 A6000  | ~ 10 |
>
> > Q3. Although I am aware that diffusion bridges can be seen as a generalization of flow matching, I am curious to hear why the authors decided opted to use diffusion bridges instead of flow matching?
>
> Thank you for this excellent question about our choice of diffusion bridges over flow matching. While both approaches can model transport between distributions, several factors made diffusion bridges more suitable for our protein surface generation task:
>
> ` 1. Stochastic exploration: `
>
> Protein-protein interactions inherently involve stochastic processes at the molecular level, including thermal fluctuations and conformational sampling. Diffusion bridges use stochastic differential equations (SDEs) that naturally incorporate noise during the generation process. This stochasticity is beneficial for exploring diverse protein surface configurations, as it allows the model to discover multiple viable binding modes rather than converging to deterministic solutions. Whereas the deterministic ordinary differential equations (ODEs) of flow matching may oversimplify the complex, probabilistic nature of surface complementarity between receptor and ligand.
>
> ` 2. Superior performance on structured biological data:`
>
> While flow matching has shown promise for image generation, diffusion models have demonstrated superior empirical performance on highly structured data like protein structures and molecular surfaces. Given that our 3D point cloud representations of protein surfaces exhibit complex geometric and biochemical constraints, the proven effectiveness of diffusion approaches on structured biological data was a decisive factor.
>
> ` 3. Flexibility in sampling: `
>
> The SDE framework allows us to easily switch between stochastic (DDPM) and deterministic (DDIM) sampling strategies, providing controllable trade-offs between diversity and quality.
>
> > Q4. It is not entirely clear to me how the authors encoded the surface structure. As I understood it, they use a point cloud for the surface, but I am not sure how many points do they use (one for each atom?) and if it could benefit from a more precise parameterization (i.e., more points).
>
> We thank the reviewer for pointing out the need for clarification. We generate solvent-accessible surface representations using PyMol, which provides a triangulated molecular surface based on a probe radius that approximates both the Solvent-Accessible Surface Area (SASA) and Solvent-Excluded Surface (SES). From this surface, we sample a point cloud in which each point is annotated with 3D spatial coordinates and relevant physicochemical features, such as hydrophobicity and hydrogen bonding potential. To encode the surface of the receptor protein, we extract node-level features from the surface points and apply an MLP to obtain embeddings. Each surface node is represented by its 3D position ($surf_t$), hydrogen bonding potential ($surf_{hbond}$), and hydrophobicity score (${surf}_{hp}$). These features are concatenated and passed through an MLP to produce the surface node embeddings.
>
> On average, we sample approximately three surface points per residue. This density was selected to balance surface resolution and computational efficiency. We also implement a subsampling strategy to avoid excessive surface points, which would significantly increase computational cost without substantial performance gains. While increasing the number of surface points could provide a finer geometric representation, our experiments showed diminishing returns in predictive performance beyond a certain point density. Exploring more adaptive or finer-grained surface parameterization remains an interesting direction for future work.
>
> -------------
> We sincerely thank you for your insightful comments, which have greatly enhanced the clarity and quality of our manuscript.

---

> ### Comment · Reviewer_WAQ2 · 2025-08-01
>
> I thank the authors for their thorough response and additional clarifications.
>
> I would like to follow up on a question regarding Q3. Did the authors consider other frameworks such as stochastic interpolants? It seems like it could also have been a good fit. Do they think that this would add any benefits?
> Also, the authors claim that diffusion models have shown superior performance. Could they provide any intuition on as to why?

---

> > ### Author Response · Authors · 2025-08-05
> > **Reply to Updated Comment**
> >
> > Thank you for your thoughtful question. We appreciate the suggestion to consider the stochastic interpolant framework.
> >
> > Stochastic interpolants provide a powerful unifying framework for several recent generative modeling approaches, including score-based diffusion models, flow matching, and diffusion bridges. These methods can be understood as learning guiding functions such as score functions, velocity fields, or drifts along a potentially stochastic path that connects a simple base distribution to the target data distribution. Specifically, score-based diffusion models rely on reverse-time stochastic differential equations or their deterministic counterparts, ordinary differential equations. Flow matching focuses on learning deterministic velocity fields to describe the transformation between source and target distributions. Diffusion bridges generalize this further by modeling conditioned paths that begin and end at specified points.
> >
> > Adopting the stochastic interpolant perspective can help unify our model within a broader generative approaches and may inspire alternative training or sampling strategies through different choices of interpolants or objectives. We will incorporate this perspective into the revised manuscript to better situate our approach in this evolving theoretical landscape.
> >
> > Due to the inherent stochasticity, diffusion models tend to be more robust when modeling complex, high-dimensional, and multimodal data manifolds. The injected noise encourages broader exploration of the data distribution’s support, helping to prevent overfitting to specific trajectories. In contrast, flow matching methods rely on deterministic transport dynamics, which—without careful regularization or training with diverse interpolants—may be more prone to overfitting or to missing modes in such spaces.
> >
> > We appreciate the reviewer’s insight, which has helped us refine both our positioning and the clarity of our claims.

---

> > > ### Comment · Reviewer_WAQ2 · 2025-08-05
> > >
> > > I thank the authors for their continued engagement and further responses to my questions. The authors have adequately responded to all my questions and have shared additional insights. I think they should add these discussions into the revised manuscript.
> > >
> > > Overall, I think that my initial assessment of the paper is still valid, and I will leave my score unchanged.

---

> > > > ### Author Response · Authors · 2025-08-05
> > > >
> > > > We appreciate your valuable input, which has helped us refine the manuscript further, and we will ensure that additional discussions are included to enhance the clarity of the paper. We’re grateful for your time and constructive feedback throughout the review process.

---

### Official Review · Reviewer_HJCF · 2025-07-03

**Clarity:** 4
**Significance:** 3
**Originality:** 3
**Rating:** 4
**Confidence:** 3

**Summary:**

This paper introduces **PepBridge**, a diffusion-based framework for joint peptide surface and structure generation conditioned on a receptor. It uses a **Surface Diffusion Bridge** to map receptor surface point clouds to complementary peptide surfaces, followed by multi-modal diffusion for backbone and side-chain modeling. A shape-frame matching module aligns surface and structure. The method shows solid performance on peptide design tasks, with the key contribution being the integration of surface geometry as a generative prior.

**Questions:**

**Questions**

1. As I understand, the model does not use receptor backbone or sequence information. Would adding these information further help?

2. Could PepFlow benefit from adding surface context similarly? Also, can you elaborate on why the performance gain is modest compared to PepFlow despite richer modeling

3. Could you also discuss the cost of surface computation, which could affect real-world applicability.

**Ethical Concerns:**

["NO or VERY MINOR ethics concerns only"]

**Limitations:**

Yes

**Quality:**

3

**Strengths And Weaknesses:**

**Strengths**

- Clarity/Quality: The paper is well-written and easy to follow. Figures and illustrations clearly convey the architecture and main ideas. The paper includes thorough ablation studies, multiple baselines, and clear evidence that each component contributes to the final performance.

- Originality: The proposed **Surface Diffusion Bridge** is a novel and elegant approach to modeling receptor-conditioned surface generation.

- Significance: The task of receptor-aware peptide generation is important and underexplored. The method highlighs the advantages of shape-driven generation in target protein design.

**Weaknesses**

- Quality: The performance improvement over strong baselines like PepFlow is relatively modest. While statistically sound, the gain may not fully justify the increased model complexity

- Significance: When surface context is removed, performance drops sharply. This raises the question that could other peptide design models, such as PepFlow, also benefit from adding surface? The core contribution may be this **surface + bridge** formulation.

---

> ### Author Rebuttal · Authors · 2025-07-30
>
> Thank you for your thoughtful questions and for engaging deeply with our work. We respond to these suggestions point-by-point below:
>
> ---------------------------
>
> > Q1. As I understand, the model does not use receptor backbone or sequence information. Would adding these information further help?
>
> Thank you for this important question. Our model does indeed incorporate both receptor backbone and sequence information.
> Specifically, for both receptor and ligand residues, we initialize node embeddings using multiple structural and sequence features: residue indices, atom coordinates, backbone dihedral angles, side-chain angles, and the diffusion timestep $t$. For edge embeddings between residue pairs, we incorporate residue-type pairs, relative sequence positions, pairwise distances, and relative orientations (details can be found in Appendix D).
>
> Our denoising network architecture is built on Invariant Point Attention (IPA), which uses SE(3)-invariant attention mechanisms to effectively model interactions between the receptor and peptide while preserving the geometric relationships encoded in the backbone structure.
>
> Our experimental results demonstrate that receptor backbone and sequence information are crucial contributors to model performance. We believe this comprehensive incorporation of structural and sequence features contributes significantly to the model's performance.
>
> > Q2. Could PepFlow benefit from adding surface context similarly? Also, can you elaborate on why the performance gain is modest compared to PepFlow despite richer modeling
>
> Thank you for this insightful question regarding both the potential for surface context in PepFlow and our performance comparison.
>
> **Surface Context in PepFlow:**
>
> PepFlow could potentially benefit from incorporating surface context similar to our approach. Surface complementarity is fundamental to protein-peptide binding, and adding explicit surface representations could enhance PepFlow's ability to model binding site compatibility and geometric constraints.
>
> **Performance Comparison and Methodological Differences:**
>
> The key differences lie in the underlying mathematical frameworks:
> PepFlow uses flow matching with deterministic ordinary differential equations (ODEs) that learn direct velocity fields between distributions. In contrast, our diffusion bridge approach employs stochastic differential equations (SDEs) that incorporate controlled noise during generation. This stochasticity offers several advantages for peptide-receptor binding:
>
>  `1. Exploration of binding diversity:`
>
> The stochastic process naturally explores multiple viable binding modes, which is crucial given that peptides often exhibit conformational flexibility at binding sites.
>
> `2. Surface complementarity modeling:`
>
> The noise inherent in SDEs helps model the probabilistic nature of surface-surface interactions, where slight variations in positioning can lead to different binding modes.
>
> `3. Sampling flexibility:`
>
> Our SDE framework allows switching between stochastic (DDPM) and deterministic (DDIM) sampling strategies, providing controllable trade-offs between diversity and binding mode exploration.
>
> **Future Directions:**
> We plan to investigate combining surface context with flow matching formulations to explore the benefits of both approaches.
>
> > Q3. Computational complexity analysis, runtime comparison with baselines, and the impact of computational efficiency on practical applicability.
>
> We appreciate the reviewer’s concern regarding the potential computational overhead introduced by our multi-modal diffusion framework. To address this, we compared PepBridge with several baseline methods in terms of training time, inference time per sample, GPU usage, and model size. The time cost is reported as the total time spent divided by the number of designed candidates.
>
> While multi-modal processing does introduce additional complexity relative to uni-modal approaches, our analysis shows that PepBridge achieves a strong balance between computational efficiency and performance. The modest increase in resource usage is well-justified by the substantial gains in structural precision and interface quality, especially in biologically critical tasks such as therapeutic protein design.
>
> ### Table: Comparison of Computation Time and Resources
>
> | Method   | Training Time  | Inference Time (s/sample) | GPU(s) Used |  Params (M) |
> |-------------------------------|---------------------|----------------------------|-------------------|------------|
> | RFdiffusion | 3 days | 80–180  | 8 A100 | ~ 120 |
> | ProteinGenerator | 4 weeks | 152 | 64 V100 | ~ 120 |
> | PepFlow | 20 hours | 14-24 | 2 A6000 | ~ 7 |
> | Chroma  | 10 weeks | 185-226 | 8 V100  | ~ 20 |
> | PepGLAD  | 29 hours | 3 | 24G GPU| ~ 2.5 |
> | PepBridge | 21 hours | 16-37 | 2 A6000  | ~ 10 |
>
> ------------------------------
> We appreciate your thorough review and advice to make our study more readable.

---

> > ### Comment · Reviewer_HJCF · 2025-08-06
> >
> > The responses are clear and appreciated. I am keeping my score unchanged. Thanks.

---

> > > ### Author Response · Authors · 2025-08-09
> > >
> > > Thank you for your feedback and for considering our responses.

---

### Official Review · Reviewer_Qs9a · 2025-07-03

**Clarity:** 3
**Significance:** 2
**Originality:** 2
**Rating:** 4
**Confidence:** 4

**Summary:**

The authors propose a joint framework to jointly design peptide sequence and structure while ensuring the surface complementarity in peptide design task. The proposed method is first use a diffusion bridge model to predict the ligand surfaces with receptor surfaces, and then jointly predict the residue type and frame structure with a multi-modal diffusion model. The authors provide a good illustration of their method and extensive experiments to show its superiority.

**Questions:**

1.There has been a long history to use surface complementarity to design ligands in traditional protein or small molecule design previous to deep learning era. I think authors should include this part of related works, and provide a discussion whether PepBridge can perform better than these methods.

2.Following the previous context, there are two questions for surface-based design:
1)How is the robustness of surface complementary methods against imperfect structure? The error of crystal structure is not significant in whole protein structure, but can be harmful when describing very local surface. If we use AlphaFold-like models to predict the receptor  structure, the error could be even larger.
2)Another problem is that the surface complementarity condition can be misleading sometimes. I think this could probably be a major advantage for learning-based methods, because they don't need to assume a very strict complementarity but can learn from data. Could authors help me understand how is PepBridge designed to address this issue?

3.I think there should be more ablation study contributed to understand the performance of separate components. For example, how is the DDBM surface diffusion part and structure design part influence each other? I notice that the authors already provide some ablation study on the change of joint performance, but I wonder if it is possible to decompose the model totally.

**Ethical Concerns:**

["NO or VERY MINOR ethics concerns only"]

**Final Justification:**

My concerns are well addressed and I think I have no further major concerns. Thus I will keep my scores unchanged as 4.

**Limitations:**

Yes

**Quality:**

3

**Strengths And Weaknesses:**

The paper is well written and easy to understand. The experiments are extensive with sufficient ablations and benchmarks. The result seems promising and proves the authors' point that the bidirectional information flow between surface and designed structure can help this task. I think a major concern is that such joint models with multiple losses and components could introduce potential problems for efficiency and robustness. However, considering the task formulation and authors' intention to jointly design surface and peptide structure, the proposed method is a reasonably sophisticated method. More ablation studies by further decomposing the model could be helpful to address this issue.

---

> ### Author Rebuttal · Authors · 2025-07-30
>
> We sincerely appreciate your valuable feedback and provide point-wise responses below.
>
> ----------------------------------------
> > Q1. There has been a long history to use surface complementarity to design ligands in traditional protein or small molecule design previous to deep learning era. I think authors should include this part of related works, and provide a discussion whether PepBridge can perform better than these methods.
>
> You are absolutely correct that surface complementarity has been a fundamental principle in protein and small molecule design well before the deep learning era. We will add a section discussing classical approaches including:
>
>
> Classical methods often relied on explicit modeling of geometric and physicochemical surface complementarity, such as shape matching or lock-and-key models. More recently, methods like MaSIF [1] represented protein surfaces as meshes and used predefined geometric and chemical descriptors to fingerprint surface patches. To reduce the high computational cost of mesh-based methods, dMaSIF [2] modeled molecular surfaces as point clouds, demonstrating competitive performance by assigning atom types and computing features directly from atomic positions.
>
> In molecular generation, ShEPhERD [3] incorporates multiple surface-level features (e.g., shape, electrostatics, and pharmacophores) into a joint diffusion-denoising framework for 3D molecular graphs. DSR [4] models protein surfaces using implicit neural representations to capture dynamic structural features over time. Similarly, SurfPro [5] generates functional protein sequences conditioned on ground-truth surfaces and their biochemical properties. However, this reliance on known target surfaces limits its applicability in scenarios where accurate surface reconstruction is difficult or unavailable.
>
> These methods demonstrate the power of surface-based design. However, they typically assume strict geometric complementarity or require hand-crafted features and ground-truth surfaces. In contrast, PepBridge leverages denoising diffusion bridge models (DDBMs) to learn a data-driven mapping between receptor and ligand interfaces. This allows PepBridge to capture non-obvious, flexible, and non-complementary interactions that classical methods might miss—especially important in peptide binding, where induced fit and flexible recognition often play a key role.
>
> We will update the manuscript to reflect this broader context and provide a more detailed comparison highlighting the strengths and limitations of surface-based methods relative to PepBridge.
>
> [1] Deciphering interaction fingerprints from protein molecular surfaces using geometric deep learning. Nature Methods.
>
> [2] Fast end-to-end learning on protein surfaces. dMaSIF CVPR 2021.
>
> [3] ShEPhERD: Diffusing shape, electrostatics, and pharmacophores for bioisosteric drug design. ICLR 2025.
>
> [4] DSR: Dynamical Surface Representation as Implicit Neural Networks for Protein. NeurIPS 2023.
>
> [5] SurfPro: Functional Protein Design Based on Continuous Surface. ICML 2024.
>
> > Q2.1 How is the robustness of surface complementary methods against imperfect structure? The error of crystal structure is not significant in whole protein structure, but can be harmful when describing very local surface. If we use AlphaFold-like models to predict the receptor structure, the error could be even larger.
>
> This is a very good question regarding the sensitivity of surface-based methods to structural inaccuracies. ! We agree that while global structural deviations may be minor (e.g., 0.5–2.0 Å RMSD), even small atomic-level perturbations can significantly affect local surface features such as curvature, pocket geometry, accessible surface area, and electrostatic potential. This challenge is especially relevant when using predicted structures from models like AlphaFold, where the overall fold is typically reliable, but side-chain placements and flexible loop regions may be less accurate.
>
> To address this issue and improve the robustness of PepBridge, we are considering the following strategies:
>
>  `1. Learned Error Tolerance:`
>
> The model could be trained on a diverse set of protein structures—including high-resolution crystal structures, NMR ensembles, and computational models—covering a broad range of structural qualities. This heterogeneity allows the model to learn representations that are inherently more tolerant to local inaccuracies and noise, improving its generalizability across different structure sources.
>
>  `2. Ensemble-Based Modeling: `
>
> To mitigate the reliance on a single static structure, especially for predicted or low-resolution inputs, PepBridge can be extended to incorporate ensemble-based inputs. Using multiple conformations or perturbed variants (e.g., from molecular dynamics simulations or AlphaFold confidence-based sampling) can help account for uncertainty and enhance prediction stability.
>
>  `3. Empirical Robustness Evaluation: `
>
> We are actively investigating robustness by evaluating PepBridge on (i) structures with synthetic coordinate perturbations, (ii) comparisons between AlphaFold-predicted structures and corresponding crystal structures; and (iii) NMR ensembles that capture native conformational variability. These experiments help quantify how sensitive predictions are to structural imperfections.
>
> Additionally, we acknowledge that proteins are not rigid entities; they undergo constant fluctuations in physiological conditions. Capturing the dynamic nature of the surface—rather than relying solely on static snapshots—could further enhance modeling accuracy for protein–protein interactions. Future work will explore integrating dynamic surface representations derived from conformational ensembles or time-averaged properties.
>
> > Q2.2 Another problem is that the surface complementarity condition can be misleading sometimes. I think this could probably be a major advantage for learning-based methods, because they don't need to assume a very strict complementarity but can learn from data. Could authors help me understand how is PepBridge designed to address this issue?
>
> Thank you for this insightful question about surface complementarity limitations. We fully agree that strict surface complementarity assumptions can be misleading and that learning-based methods offer significant advantages in this regard.
>
> However, it is worth noting that several key innovations are specifically designed in PepBridge to address the limitations of rigid surface complementarity:
>
>  `1. Learning-based Surface Relationship Modeling: `
>
> Rather than assuming strict geometric complementarity, PepBridge employs denoising diffusion bridge models (DDBMs) to learn the complex relationship between receptor and ligand surfaces from data. This allows the model to capture subtle, non-complementary interactions that are crucial for binding but would be missed by purely geometric approaches.
>
>  `2. Integration of Biochemical Properties: `
>
> PepBridge goes beyond surface geometry by incorporating biochemical properties alongside geometric features. This enables the model to learn that effective binding often involves electrostatic interactions, hydrophobic patches, and other physicochemical factors that don't necessarily require perfect shape complementarity.
>
>  `3. Flexible Surface-Structure Coupling: `
>
>  Our joint design framework enables dynamic adjustment between surface geometry and backbone structure through Shape-Frame Matching Networks (SFMN). This flexibility encourages the generation of binding interfaces that may not exhibit perfect complementarity but are biologically relevant and energetically favorable.
>
>  `4. Multi-Step Refinement Process: `
>
>  The multi-model diffusion approach allows for iterative refinement where initial surface predictions can be adjusted based on structural constraints and vice versa, leading to more realistic binding interfaces that balance complementarity with other binding determinants.
>
> All above designs make PepBridge generate peptides that engage targets through diverse binding modes, including those involving conformational flexibility and partial complementarity - scenarios where traditional geometric complementarity assumptions would fail.
>
> > Q3. I think there should be more ablation study contributed to understand the performance of separate components. For example, how is the DDBM surface diffusion part and structure design part influence each other? I notice that the authors already provide some ablation study on the change of joint performance, but I wonder if it is possible to decompose the model totally.
>
> We thank the reviewer for this insightful question. To better understand the contributions of individual components in our model, we performed ablation studies using several PepBridge variants with and without surface diffusion information. As shown in the table below, incorporating surface features improves performance across the evaluation metrics.
>
> | | Div\_stru (↑) | Aff. % (↑) | Stab. % (↑) | RMSD Å (↓) | BSR (↑)|
> |:--|:-:|:----:|:---:|:---:|:----:|
> |PepBridge w/Bb+Seq         |0.56 |17.12  |16.41|2.87|79.79|
> |PepBridge w/Bb+Seq+surf    |0.62 |22.07  |20.71|2.18|84.91|
> |PepBridge w/Bb+Seq+Ang     |0.53 |18.46  |19.73|3.21|76.53|
> |PepBridge w/Bb+Seq+Ang+surf|0.59 |19.16  |25.02|2.19|83.90|
>
> -----------------------------
>
> We would like to thank you again for your valuable comment and suggestions, which significantly improve our paper quality.

---

> ### Comment · Reviewer_Qs9a · 2025-08-09
>
> This answer is adequate and has addressed all my concerns. I especially appreciate your detailed ablation study. I will keep my scores unchanged.

---

> > ### Author Response · Authors · 2025-08-09
> >
> > Thank you for your feedback and for helping us improve the paper.

---

### Note · Authors · 2025-08-14

We thank the reviewers for their constructive feedback, which has significantly strengthened our work. Below, we summarize the key suggestions and our corresponding updates.

1. **Baseline Models (Q8ka)**:

We added two additional baselines PepHAR and PPFLOW to broaden the scope of our comparative analysis.

2. **More Discussion (Qs9a, WAQ2)**:

In response to requests for deeper discussion on surface complementarity in ligand design, we have added coverage of both classical approaches and recent deep learning–based methods.
We also expanded our explanation of how the stochastic interpolant framework relates to both diffusion bridges and flow matching, clarifying the conceptual and methodological links between these paradigms.

3. **Additional Results (Qs9a, WAQ2, Q8ka)**:

We updated our results to include:
- Additional ablation studies isolating the effects of individual model components.
- Hyperparameter sensitivity analysis, focusing on the impact of sampling steps.
- Further analysis of protein surface characteristics.

4. **Computational Efficiency (HJCF, WAQ2, Q8ka)**:

We added a table comparing PepBridge with multiple baselines in terms of training time, inference time per sample, GPU usage, and parameter count.

---

### Decision · Program_Chairs · 2025-09-17

**Decision:**

Accept (poster)

**Comment:**

This submission introduces a joint framework for designing peptide sequences and structures while explicitly ensuring surface complementarity. The proposed surface diffusion bridge for receptor-conditioned surface generation is considered both novel and elegant. By adding baseline models, improving computational efficiency, and expanding the discussion, the authors have strengthened the robustness and readability of the paper. Reviewer Q8ka raised concerns about hyperparameters—particularly the time step—and aspects of the experimental setup. These points should be addressed in future revisions in order to further improve evaluation rigor. Overall, I recommend acceptance of this paper.